# The autophagic membrane tether ATG2A transfers lipids between membranes

**Shintaro Maeda, Chinatsu Otomo, Takanori Otomo\***

Department of Integrative Structural and Computational Biology, The Scripps Research Institute, La Jolla, United States

**Abstract** An enigmatic step in de novo formation of the autophagosome membrane compartment is the expansion of the precursor membrane phagophore, which requires the acquisition of lipids to serve as building blocks. Autophagy-related 2 (ATG2), the rod-shaped protein that tethers phosphatidylinositol 3-phosphate (PI3P)-enriched phagophores to the endoplasmic reticulum (ER), is suggested to be essential for phagophore expansion, but the underlying mechanism remains unclear. Here, we demonstrate that human ATG2A is a lipid transfer protein. ATG2A can extract lipids from membrane vesicles and unload them to other vesicles. Lipid transfer by ATG2A is more efficient between tethered vesicles than between untethered vesicles. The PI3P effectors WIPI4 and WIPI1 associate ATG2A stably to PI3P-containing vesicles, thereby facilitating ATG2A-mediated tethering and lipid transfer between PI3P-containing vesicles and PI3P-free vesicles. Based on these results, we propose that ATG2-mediated transfer of lipids from the ER to the phagophore enables phagophore expansion.
DOI: https://doi.org/10.7554/eLife.45777.001

## Introduction

Autophagy is the bulk degradation-recycling process that plays crucial roles in the maintenance of cellular homeostasis in eukaryotes (*Choi et al., 2013*; *Levine and Kroemer, 2019*; *Mizushima and Komatsu, 2011*; *Reggiori and Klionsky, 2013*). Upon the induction of autophagy, a portion of the cytoplasm is sequestered within the double-membraned autophagosome compartment and transported to the lysosome (*Lamb et al., 2013*; *Mizushima et al., 2011*). Autophagosome biogenesis begins with the nucleation of the phagophore (also called the isolation membrane) adjacent to the endoplasmic reticulum (ER), followed by the expansion of the phagophore into a large cup-shaped double-membraned structure, resulting in the engulfment of bulk cytoplasmic constituents of various sizes ranging from protein molecules to organelles. Notably, the edge of the cup-shaped phagophore is associated with the ER (*Hayashi-Nishino et al., 2009*; *Uemura et al., 2014*; *Ylä-Anttila et al., 2009*), and this association is maintained during phagophore expansion (*Graef et al., 2013*; *Suzuki et al., 2013*). This intimate spatial relationship has led to the hypothesis that the ER feeds the phagophore with lipids, enabling phagophore expansion (*Tooze and Yoshimori, 2010*). The phagophore is enriched with the lipid molecule phosphatidylinositol 3-phosphate (PI3P) (*Cheng et al., 2014*; *Obara et al., 2008a*), whose role is to recruit the PROPPIN family PI3P effectors Atg18/WD-repeat protein interacting with phosphoinositides (WIPIs) (Atg18 in yeast and WIPI1-4 in mammals) (*Barth et al., 2001*; *Dove et al., 2004*; *Mercer et al., 2018*; *Proikas-Cezanne et al., 2015*; *Proikas-Cezanne et al., 2004*) and the Atg18/WIPI-binding protein autophagy-related 2 (ATG2) (Atg2 in yeast and ATG2A/B in mammals) (*Mizushima et al., 2011*; *Obara et al., 2008b*; *Shintani et al., 2001*; *Velikkakath et al., 2012*; *Wang et al., 2001*). Yeast studies have shown that both Atg18 and Atg2 localize exclusively to the phagophore edge and are required for phagophore expansion (*Gómez-Sánchez et al., 2018*; *Graef et al., 2013*; *Suzuki et al., 2013*), and mammalian studies have echoed the importance of ATG2A/B in phagophore expansion (*Kishi-Itakura et al.,*

**\*For correspondence:**
totomo@scripps.edu

**Competing interests:** The authors declare that no competing interests exist.

*2014*; *Tamura et al., 2017*; *Velikkakath et al., 2012*). However, the mechanism through which these proteins enable membrane expansion remains unknown (*Mizushima, 2018*).

We recently characterized the structural and biochemical properties of the human ATG2A-WIPI4 complex (*Chowdhury et al., 2018*). The 1938 residue-long ATG2A is folded into a 20 nm-long, 30 Å-wide rod with both tips composed of Pfam database-registered conserved domains: a 'Chorein_N' domain at the N-terminus of one tip (referred to as the N tip) and an 'ATG2_CAD' domain in the middle of the sequence of the other tip (referred to as the CAD tip) (*Figure 1A*). Chorein is one of vacuolar protein sorting 13 (VPS13) family proteins that function at contact sites between various organelles (*Kumar et al., 2018*; *Lang et al., 2015*; *Murley and Nunnari, 2016*; *Park et al., 2016*); as the name suggests, the Chorein_N domain is found at the N-terminus of VPS13 proteins (*Pfisterer et al., 2014*; *Velayos-Baeza et al., 2004*). The ATG2_CAD domain is unique to ATG2, and WIPI4 is tightly bound adjacent to the CAD tip (*Figure 1A*). The yeast Atg2-Atg18 and the rat ATG2B-WIPI4 complexes exhibit similar overall shapes, suggesting that the structure is evolutionarily conserved and, therefore, likely important for the function of these complexes (*Chowdhury et al., 2018*; *Zheng et al., 2017a*). Indeed, each tip of the ATG2A rod can bind to a membrane, and simultaneous membrane binding of both tips results in tethering the two membranes ~ 10–20 nm apart (*Chowdhury et al., 2018*), the typical distance observed at various organelle contact sites (*Gatta and Levine, 2017*). ATG2 proteins rely on lipid-packing defects in the membrane bilayer to effect membrane binding: Atg2 associates with giant unilamellar vesicles containing phosphatidyl-ethanolamine (PE), a lipid molecule with a small head group that introduces packing defects

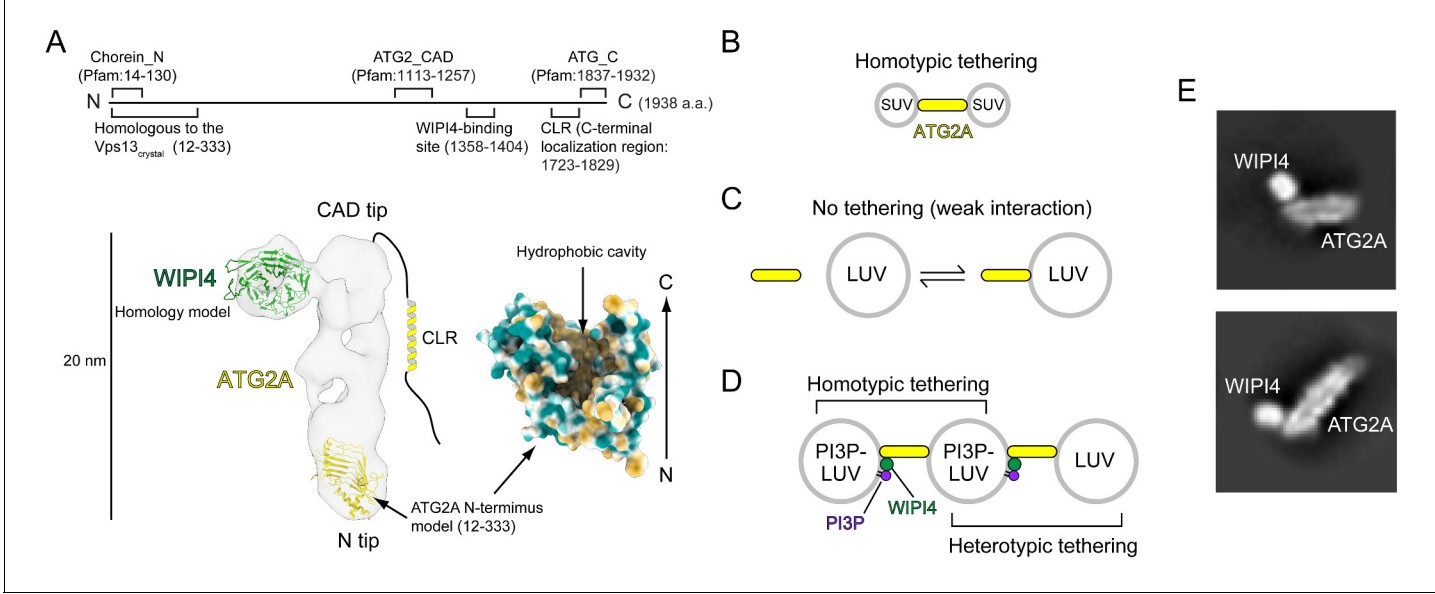

**Figure 1.** Structural and biochemical properties of the ATG2A-WIPI4 complex. (**A**) Summary of previous structural characterizations of the ATG2A-WIPI4 complex. The conserved and functional regions of ATG2A are indicated on the line-represented primary structure (top). The negative stain EM map (EMD-8899) of the ATG2A-WIPI4 complex is shown with docked homology models of WIPI4 and the ATG2A N-terminus as well as additional information regarding the C-terminal regions. The homology models were obtained from the I-TASSER server (*Roy et al., 2010*), and the molecular model was generated using ChimeraX (*Goddard et al., 2018*). The surface representation of the ATG2A N-terminus model is colored according to the hydrophobicity potential calculated in ChimeraX. Hydrophobic and hydrophilic residues are shown in khaki and turquoise, respectively. (**B–D**) Summary of the membrane binding/tethering properties of ATG2A and the ATG2A-WIPI4 complex reported previously (*Chowdhury et al., 2018*). (**B**) ATG2A binds tightly to and tethers SUVs. (**C**) ATG2A associates weakly with and does not tether LUVs. (**D**) The ATG2A-WIPI4 complex can tether PI3P-containing LUVs and tether a PI3P-containing LUV to a PI3P-free LUV. Note that the tethering events shown in (**B**) and (**D**) lead to vesicle clustering in test tubes, but, for clarity, only one pair of tethered vesicles for each tethering pattern is shown. (**E**) Cryo-EM 2D class averages of the ATG2A-WIPI4 complex.

DOI: https://doi.org/10.7554/eLife.45777.002

The following figure supplement is available for figure 1:

**Figure supplement 1.** Representative cryo-EM image of the ATG2A-WIPI4 complex.

DOI: https://doi.org/10.7554/eLife.45777.003

(*Gómez-Sánchez et al., 2018*), as well as small unilamellar vesicles (SUVs) (*Kotani et al., 2018*), whose high curvatures create lipid-packing defects (*Harayama and Riezman, 2018*). ATG2A also associates tightly with SUVs but only weakly with large unilamellar vesicles (LUVs) (*Chowdhury et al., 2018*), whose lower curvatures create fewer lipid-packing defects. In correlation with the strength of the interactions, ATG2A/Atg2 tethers SUVs (*Figure 1B*) (*Chowdhury et al., 2018*; *Kotani et al., 2018*), while ATG2A cannot tether LUVs (*Figure 1C*) (*Chowdhury et al., 2018*). However, the ATG2A-WIPI4 complex can mediate homotypic tethering between two PI3P-containing LUVs as well as heterotypic tethering between a PI3P-containing LUV and a PI3P-free LUV (*Figure 1D*) (*Chowdhury et al., 2018*), presumably because the N tip has affinity to LUVs and WIPI4 can direct the CAD tip to the PI3P-containing LUV. These data led us to propose that the ATG2A-WIPI4 complex tethers PI3P-positive phagophores to neighboring membranes, such as the ER and vesicles (*Chowdhury et al., 2018*). This model is supported by yeast studies concluding that the Atg2-Atg18 complex mediates the ER-phagophore association (*Gómez-Sánchez et al., 2018*; *Kotani et al., 2018*).

ATG2 contains two additional conserved regions: an 'ATG_C' domain, also registered in Pfam and located at the end of the C-terminus; and another short region preceding the ATG_C domain (*Figure 1A*). Despite the name, ATG_C domains are also found in VPS13 proteins and are responsible for the localization of both ATG2A/B and VPS13 proteins to lipid droplets (*Kumar et al., 2018*; *Tamura et al., 2017*). Consistently, the ATG_C domain of ATG2A/B is dispensable for autophagy (*Tamura et al., 2017*). In contrast, the region preceding the ATG_C domain is required for the localization of ATG2A/B and Atg2 to phagophores (*Kotani et al., 2018*; *Tamura et al., 2017*; *Velikkakath et al., 2012*). This region, referred to as the C-terminal localization region (CLR), contains amphipathic α-helices that can associate with membranes (*Chowdhury et al., 2018*; *Kotani et al., 2018*; *Tamura et al., 2017*). The CLR of yeast Atg2 is required for the membrane tethering activity of this protein in vitro (*Kotani et al., 2018*) but is not required for that of ATG2A, consistent with the structural observation that the rod is the membrane tethering unit of ATG2A (*Chowdhury et al., 2018*). However, no rational explanations have been provided to clarify this discrepancy. Previous structural studies could not identify the location of the ATG2A/B C-terminus in the electron microscopy (EM) maps, suggesting that the C-terminus is flexible (*Chowdhury et al., 2018*; *Zheng et al., 2017a*). Such flexibility may be advantageous for the C-terminal regions to function sufficiently as localization determinants (*Kotani et al., 2018*; *Tamura et al., 2017*; *Velikkakath et al., 2012*).

These recent advancements defined the role of ATG2-Atg18/WIPI4 complexes as mediators of the ER-phagophore edge association but did not explain the means by which such membrane associations lead to phagophore expansion. The size, overall shape, and membrane tethering activity of ATG2A called to mind tubular lipid transfer proteins, such as the extended synaptotagmins and the ER-mitochondrial encounter structure (ERMES) complex (*AhYoung et al., 2015*; *Jeong et al., 2017*; *Schauder et al., 2014*; *Wong et al., 2019*), which further led us to hypothesize that ATG2A could be a lipid transfer protein. This hypothesis is strongly supported by a recent study reporting that VPS13 family proteins are lipid transfer proteins (*Kumar et al., 2018*). The crystal structure of the N-terminal 335-residue-long fragment of *Chaetomium thermophilum* Vps13p revealed a unique structure with a large hydrophobic cavity that was suggested to accommodate lipid molecules. An N-terminal 1350-residue-long construct of *S. cerevisiae* Vps13p termed Vps13α was shown to bind an unusually large number (~10) of glycerolipid molecules with a broad specificity and transfer those lipids between membranes. Homology modeling suggests that the structure of the ATG2 N-terminus would be very similar to that of Vps13p (*Figure 1A*). The VPS13 structure and the ATG2A structure proposed by the homology model fit into the N tip of our previous negative stain EM density map of ATG2A and occupy ~20–25% of the total volume (*Figure 1A*) (*Otomo et al., 2018*). The central β-sheet, which serves as the base of this structure, is expected to extend in the C-terminal direction in the full-length protein (*Kumar et al., 2018*), which would create an elongated groove-shaped hydrophobic cavity in the ATG2A rod. Such a cavity could accommodate a large number of lipids, as previously reported for VPS13 (*Kumar et al., 2018*).

Lipid transfer by ATG2 could be the fundamental molecular mechanism underlying phagophore expansion. Therefore, we investigated whether ATG2 is a lipid transfer protein by performing a series of lipid transfer assays with ATG2A. Our strategy for this study was based on the membrane binding/tethering properties of ATG2A described above (*Figure 1B–1D*). Our new results described

below collectively show that ATG2A can indeed transfer lipids between membranes. WIPI4/WIPI1-enabled tethering facilitates ATG2A-mediated lipid transfer between a PI3P-containing membrane and a PI3P-free membrane. The demonstration of lipid transfer between membranes recapitulating a PI3P-positive phagophore and the ER allows us to propose that ATG2A-mediated lipid transfer drives phagophore expansion.

## Results

### ATG2A extracts lipids from membranes and unloads the lipids to membranes

We performed a cryo-EM single-particle analysis to gain further structural information on the ATG2A-WIPI4 complex in a stain-free environment (*Figure 1—figure supplement 1*) and obtained 2D class averages that are overall consistent with our previous negative stain EM data (*Figure 1E*). These 2D images indicated that the specimen was not of sufficient quality for high-resolution structure determination but could visualize the predicted internal cavity of ATG2A, which appears to extend from the N tip to the CAD tip, supporting the hypothesis that ATG2A may be a lipid-binding/transfer protein.

Lipid transfer proteins can extract lipid molecules from existing membranes (*Wong et al., 2019*). To examine whether ATG2A has such activity, we adapted a lipid extraction assay in which liposomes containing fluorescent lipids (nitrobenzoxadiazole-conjugated phosphatidylethanolamine: NBD-PE) are incubated with proteins, followed by liposome flotation to isolate the lipid-bound proteins (*Kawano et al., 2018*). To isolate ATG2A efficiently, we used LUVs prepared by extrusion through a filter with a pore size of 100 nm, because ATG2A would interact only weakly with those LUVs (*Figure 1C*). First, ATG2A was incubated with LUVs containing 20% NBD-PE and 80% 1,2-dioleoyl (DO)-phosphatidylcholine (DOPC). This incubation did not alter the overall shape and size of the liposomes as monitored by dynamic light scattering (DLS) and negative stain EM (*Figure 2—figure supplements 1* and *2*). Then, the proteins and LUVs were separated by Nycodenz gradient-based liposome flotation (*Figure 2A*). Upon centrifugation, the fluorescence signal of the protein-free control migrated from the bottom to the top of the tube, confirming that the LUVs floated (*Figure 2B and C*). Similarly, the top fraction of the sample containing ATG2A became fluorescent, but the bottom remained so as well (*Figure 2B and C*). SDS-PAGE analysis showed that most of the ATG2A proteins were in the bottom fraction (*Figure 2D*), confirming that ATG2A did not bind tightly to these LUVs. Native PAGE analysis of the bottom fractions showed that the protein band was fluorescent (*Figure 2E*) and negative stain EM showed no liposomes in the bottom fractions (*Figure 2—figure supplement 2*), which together indicate that the proteins in the bottom fraction were bound to NBD-PE. Collectively, these data demonstrate that ATG2A can extract NBD-PE from membranes and dissociate from the membrane with those lipids attached.

We next tested the unloading of fluorescent lipids from the protein. The bottom fractions from the extraction assay described above were collected and incubated with nonfluorescent LUVs (100% DOPC) or buffer as the control and were then resubjected to liposome flotation (*Figure 2F*). After centrifugation, the fluorescence signal of the control remained in the bottom fraction, whereas that of the sample containing LUVs was detected not only in the bottom but also the top fraction (*Figure 2G and H*). In both samples, the ATG2A proteins remained in the bottom fraction (*Figure 2I*). These data indicate that ATG2A unloads NBD-PE onto LUVs during a transient interaction with the LUVs. The bottom fraction of the sample with LUVs yielded less (~33%) fluorescence signals than that of the LUV-free control. While this indicates that a substantial amount of lipids remained bound to the proteins, the signal reduction confirms the unloading of NBD-PE. In conclusion, the lipid extraction and unloading capabilities of ATG2A demonstrated here define ATG2A as a lipid transfer protein.

### ATG2A transfers lipids between tethered membranes

The inefficient unloading observed above led us to hypothesize that tethering membranes by ATG2A could facilitate lipid transfer. We first examined whether lipids can be transferred between membranes tethered by ATG2A. To this end, we turned to our previous finding that ATG2A tethers SUVs efficiently (*Figure 1B*) and performed a kinetic fluorescence lipid transfer assay with ATG2A

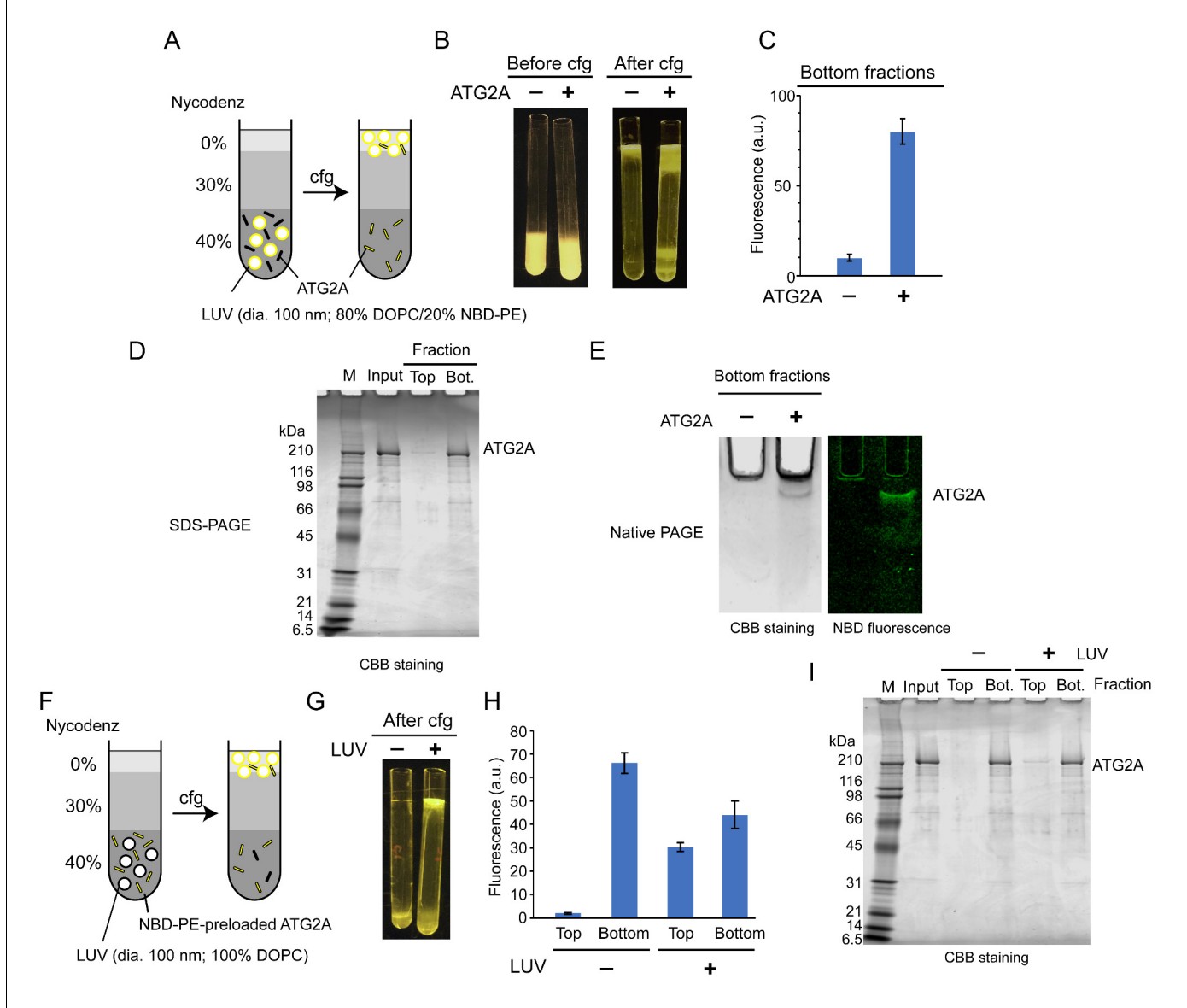

**Figure 2.** ATG2A extracts fluorescent lipids from membranes and unloads the lipids onto membranes. (A–E) NBD-PE extraction assay. (A) Diagram of the assay based on density (Nycodenz) gradient centrifugation. LUVs were prepared by extrusion through a 100 nm filter. (B) Pre and postcentrifugation fluorescence images of the centrifuge tubes with and without ATG2A. (C) Quantification of the fluorescence signal in the bottom fractions of the tubes shown in (B). (D) SDS-PAGE (4–20% acrylamide gradient gel) with Coomassie blue (CBB) staining of the bottom fractions in (B). (E) Native PAGE (4–10% acrylamide gel) of the bottom fractions in (B). The nonionic detergent decyl maltoside (0.2%) was added to the samples to prevent protein aggregation in the wells of the gel. CBB staining (top) and fluorescence (bottom) images are shown. (F–I) NBD-PE unloading assay. (F) Diagram of the assay. NBD-PE-preloaded ATG2A obtained from the bottom fraction in (B) was mixed with nonfluorescent LUVs (100 nm, 100% DOPC) and subjected to density gradient centrifugation. (G) A postcentrifugation fluorescence image of the centrifuge tubes with and without LUVs. (H) Fluorescence signal in the fractions in (G). (I) SDS-PAGE (4–20% acrylamide gradient gel) with CBB staining of the fractions in (G). Experiments were repeated three times. The fluorescence data in (C) and (H) and are shown as the average of the three repeats with the SD.

DOI: https://doi.org/10.7554/eLife.45777.004

The following figure supplements are available for figure 2:

**Figure supplement 1.** Lipid extraction by ATG2A does not affect the size of the substrate liposomes.
DOI: https://doi.org/10.7554/eLife.45777.005

**Figure supplement 2.** Lipid extraction by ATG2A does not deform LUVs, and no liposomes remain in the bottom fraction after liposome flotation centrifugation.
DOI: https://doi.org/10.7554/eLife.45777.006

and SUVs (*Figure 3A*). In this accepted assay (*Struck et al., 1981*), two kinds of vesicles are prepared and used as a mixture: donor vesicles that contain a pair of fluorescent lipids—NBD-PE and Rhodamine-PE (Rh-PE)—and acceptor vesicles that contain neither of the fluorescent lipids. Initially, NBD fluorescence arises solely from the donor vesicles and is suppressed due to quenching by Rhodamine on the same vesicle. However, owing to the dilution, NBD fluorescence increases upon the transfer of NBD-PE or Rh-PE or both NBD-PE and Rh-PE to the acceptor vesicles. To stabilize the ATG2A-SUV association, we prepared donor and acceptor SUVs by sonication and included 25% DOPE and 25% DO-phosphatidylserine (DOPS) in both vesicles. The role of PE was described above, and DOPS improves the ATG2A-membrane association (*Chowdhury et al., 2018*). As shown in *Figure 3B*, NBD fluorescence increased in the presence but not the absence of ATG2A proteins. A control experiment performed with no acceptor SUVs yielded no increase in NBD fluorescence. Thus, the signal increase observed in the presence of both SUVs resulted from the transfer of NBD-PE from the donor SUVs to the acceptor SUVs but not from the solubilization of NBD-PE by the proteins. Note that the increase in NBD fluorescence could also be explained by hemifusion or fusion between the donor and acceptor vesicles. However, these possibilities are ruled out by our previous data showing that ATG2A-clustered liposomes are disassembled upon protease treatment (*Chowdhury et al., 2018*). To further confirm this finding, we added sodium dithionite to the postreaction mixtures, which resulted in a decrease in NBD fluorescence in both samples—with and without protein—to the same level (~50% of the initial signal) (*Figure 3B*). As explained in *Figure 3A*, these results indicate that fusion did not occur.

We next performed a titration of ATG2A. ATG2A facilitated lipid transfer in a concentration-dependent manner (*Figure 3C*), suggesting that the reaction was catalyzed by ATG2A. Notably, the plateau increased as the protein concentration increased. The limited dynamic range of each experiment indicates that only a fraction of the SUVs served as substrates for a given concentration of ATG2A, which could be explained by the fact that once ATG2A became engaged with two vesicles, the ATG2A acted only on this pair. However, the protein concentrations were higher, even at the lowest concentration tested (6 nM), than that of the liposomes, which was estimated to be ~2–5 nM of each donor and acceptor with a lipid concentration of 25 μM assuming that a liposome with a radius of ~10–20 nm is composed of ~5000–15000 lipids. Thus, under all experimental conditions, most substrate SUVs should be bound by ATG2A. A simple explanation to resolve this discrepancy is that only a fraction of the proteins was actually active. Effective concentrations of ATG2A below the liposome concentration would limit the dynamic range of the fluorescence signal if there was no turnover, as mentioned above. Alternatively, ATG2A molecules as a whole might be engaged with only a fraction of substrate SUVs. This situation is likely because there would probably be a positive feedback loop in ATG2A-induced vesicle clustering: a pair of ATG2A-pretethered SUVs would serve as a more favorable binding site for another ATG2A molecule compared to a yet-untethered single SUV, as the two membranes of the pair have already been spaced optimally for the second ATG2A molecule. Hence, each vesicle in a cluster would be tethered to another vesicle by multiple ATG2A molecules, and such multivalent tethering would stabilize the vesicle cluster. This alternative explanation is supported by the observation that the NBD-PE transfer rate (per protein molecule) decreased as the protein concentration increased (*Figure 3D*). Under the first scenario (the effective protein concentration is low), the transfer rate should be maintained even at high protein concentrations, but under the second one, the NBD-PE molecules in each liposome could become limiting for multiple ATG2A molecules bound onto each liposome. The fastest rate of NBD-PE transfer was ~0.017 $s^{-1}$ per protein molecule, which was observed at the two lowest protein concentrations (6 and 12 nM). This transfer rate is comparable to the NBD-PE transfer rate of the Mmm1-Mdm12 complex, the lipid transfer machinery that exchanges PE and PS between ER and mitochondria (*Kawano et al., 2018*), but it is much slower than the rate of cholesterol transfer by OSBP between ER and Golgi (~1 $s^{-1}$ per protein molecule)(*Mesmin et al., 2013*). Note that the actual lipid transfer rate by ATG2A could be faster if ATG2A transfers not only NBD-PE but also other lipids. Furthermore, the lipid transfer rates obtained from this FRET-based method should be understood with care; those rates are likely inaccurate if Rh-PE and other lipids were also being transferred, which would complicate the quantitative analysis and is likely the case with ATG2A. As such, the details will need to be verified. Qualitatively, however, our titration data suggest that ATG2A transfers NBD-PE between tethered membranes.

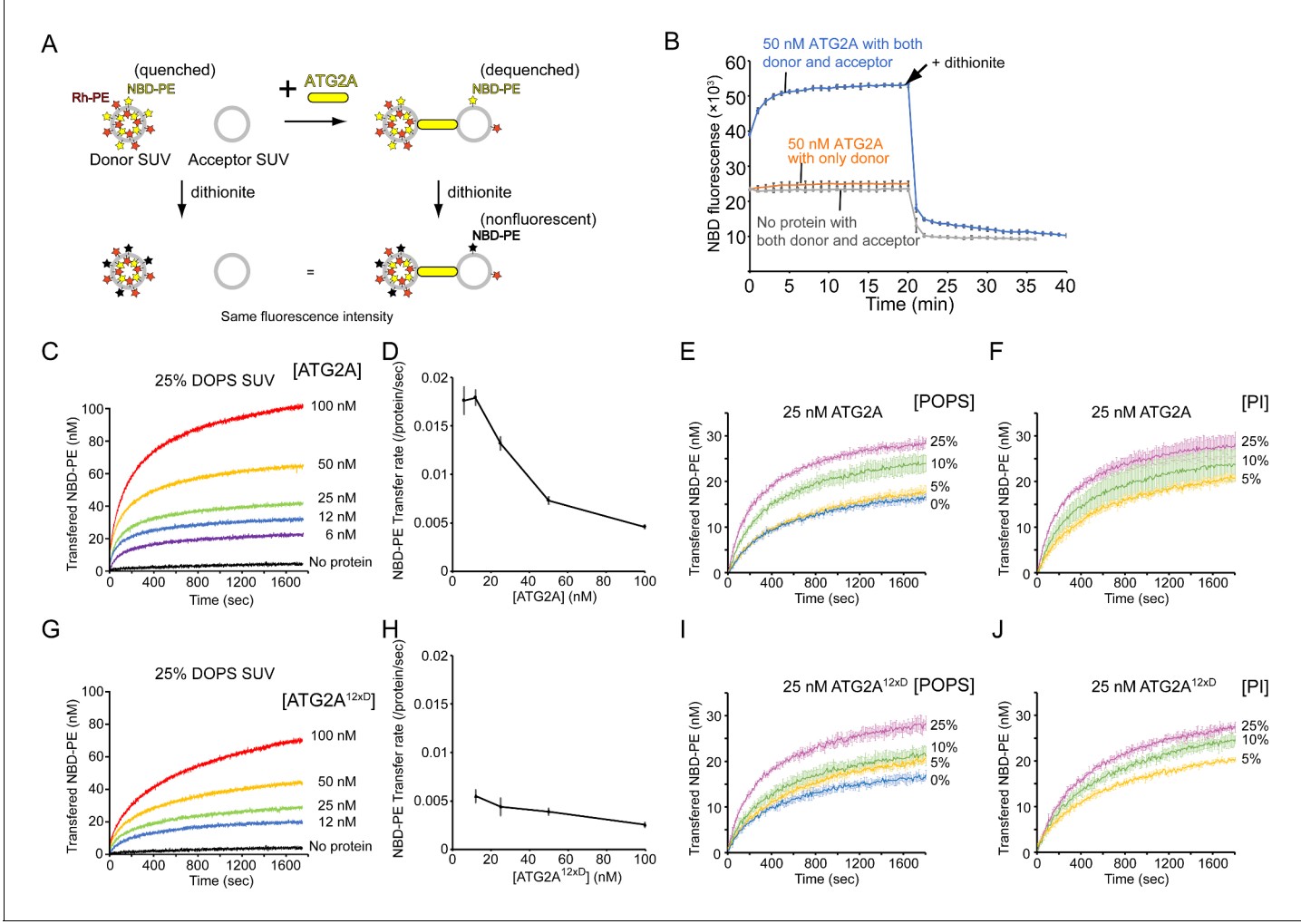

**Figure 3.** ATG2A transfers lipids between SUVs. (**A**) Diagram of the NBD fluorescence-based lipid transfer assay with ATG2A and SUVs. NBD fluorescence increases upon dilution of NBD, which is interpreted as the transfer of NBD-PE from the donor to the acceptor. Addition of dithionite would cause the loss of NBD fluorescence from the outer leaflets of liposomes (***Meers et al., 2000***). If the donor and the acceptor vesicles fuse, NBD-PE on the inner leaflet is diluted, thereby contributing to the signal increase. Hence, dithionite treatment increases the fluorescence of a membrane-fused sample to a higher level than that of a nonfused sample. (**B**) Lipid transfer assay with 50 nM ATG2A and 25 μM (lipid concentration) each of the donor SUVs (46% DOPC, 25% DOPE, 25% DOPS, 2% NBD-PE, and 2% Rh-PE) and the acceptor SUVs (50% DOPC, 25% DOPE, and 25% DOPS). SUVs were prepared by sonication, and the experiments were performed at 30 ˚C. Each data point is presented as the average of the three independent experiments and shown with the SD. Sodium dithionite was added at the time indicated by the arrow. (**C**) Titration of the ATG2A protein in a lipid transfer assay with the same donor and acceptor vesicles as in (**B**). The experiments were performed at 25 ˚C and repeated three times. A representative data set is shown. The concentrations of ATG2A are indicated. NBD fluorescence was normalized to the maximum NBD fluorescence, which was measured upon the addition of 0.1% Triton X-100 at the end of each experiment, and converted to the concentration of the transferred NBD-PE as reported previously (***Kawano et al., 2018***). (**D**) Plot of the initial NBD-PE transfer rate versus the concentration of ATG2A of the data shown in (**C**). The SD of each time point was obtained from three repeats. (**E**) Lipid transfer assay with 25 nM ATG2A and 25 μM donor SUVs containing 0, 5, 10, or 25% POPS (the other lipid components: 25% POPE, 2% NBD-PE, 2% Rh-PE, and 71, 66, 61, or 46% POPC, respectively) and 25 μM acceptor SUVs containing the same amount of POPS as the donor (the lipid compositions: 25% POPE, 0, 5, 10, or 25% POPS, and 75, 70, 65, or 50% POPC, respectively) (n = 3). The percentage of POPC of each liposome was calculated as 100% minus the fractions of all the other lipids. (**F**) The same experiment as (**E**), except that POPS was replaced by bovine liver PI (n = 3). (**G–J**) The same set of experiments (G, I, and J) and analyses (**H**) on ATG2A$^{12\times D}$ as shown in (**C–F**).

DOI: https://doi.org/10.7554/eLife.45777.007

The following figure supplement is available for figure 3:

**Figure supplement 1.** ATG2A clusters SUVs composed of PO lipids.
DOI: https://doi.org/10.7554/eLife.45777.008

In the above-described experiments, we used liposomes consisting of DO lipids, which are composed of two monounsaturated acyl chains, including a high percentage (25%) of the negatively charged DOPS, but most ER lipids are composed of a mixed saturated/unsaturated acyl chain, and negatively charged lipids are not abundant in the ER membrane (*Bigay and Antonny, 2012*; *van Meer et al., 2008*). To clarify the lipid transfer activity of ATG2A in more physiological conditions, we tested ATG2A-mediated lipid transfer with SUVs consisting of 1-palmitoyl, 2-oleoyl (PO) lipids with varying percentages of POPS (25% POPE, 0, 5, 10, or 25% POPS, and POPC). Lipid transfer between 25% POPS-containing PO SUVs was slower than that between 25% DOPS-containing DO SUVs (*Figure 3C and E*), and the more POPS the SUV contained, the faster the lipid transfer was. There was approximately a twofold difference in the transfer rate between 25% and 0% POPS (*Figure 3E*). We performed DLS experiments to examine tethering of these PO SUVs and confirmed that all SUVs clustered in the presence of ATG2A (*Figure 3—figure supplement 1*). The 25% POPS-containing SUVs clustered into the largest assembly and the other three SUVs with 0, 5, or 10% POPS clustered into a similar size, as suggested by the size distribution profiles (*Figure 3—figure supplement 1A*), while the autocorrelation profiles indicated that the liposomes with more POPS in the 0–10% range clustered into slightly larger assemblies (*Figure 3—figure supplement 1B*). Collectively, these data suggest that POPS facilitates both membrane tethering and lipid transfer. The effect of POPS appears to be based on its negative charge because the replacement of POPS with another negatively charged lipid, phosphatidylinositol (PI), yielded a similar trend in the lipid transfer assay (*Figure 3F*). Thus, the lipid transfer activity of ATG2A shares a similar preference with the membrane tethering activity: membranes rich in lipid-packing defects and negative charges are better substrates for both activities. While this finding could suggest that ATG2A may not be operating at its full capacity in cells, these experiments with PO lipids demonstrated that ATG2A can mediate lipid transfer between membranes with near-physiological lipid compositions.

We also tested a mutant protein, referred to as ATG2A[12×D], in which twelve hydrophobic residues in the amphipathic helices in the CLR are mutated to aspartic acid (*Chowdhury et al., 2018*). These mutations diminish the membrane-binding capacity of the CLR, and similar mutations impair autophagy (*Tamura et al., 2017*). Thus, ATG2A[12×D] was expected to be incompetent in membrane tethering. However, our previous results indicated that the mutant can tether membranes in vitro (*Chowdhury et al., 2018*). As shown in *Figure 3G*, ATG2A[12×D] exhibited lipid transfer activity between DO SUVs, but its transfer rate was only ~1/3 of that of the wild-type protein (*Figure 3D and H*). However, the lipid transfer between PO SUVs was only slightly slower compared with the same experiment with the wild-type protein (*Figures 3E, F, I and J*). Thus, it appears that the CLR and non-physiological DO lipids including the high percentage of DOPS synergized in promoting tethering and lipid transfer by the wild-type protein, but in more physiological PO lipids-based conditions, the CLR contributes much less to lipid transfer and perhaps to membrane tethering as well. Together with our previous data indicating that the rod-shaped part of ATG2A tethers membranes and the CLR is dispensable for membrane tethering (*Chowdhury et al., 2018*), the presented results confirm that the rod region of the ATG2A protein is the core catalytic domain for both membrane tethering and lipid transfer.

## WIPI4 facilitates ATG2A-mediated lipid transfer

Next, we sought to test our hypothesis that membrane tethering facilitates lipid transfer. To compare lipid transfer between tethered membranes and nontethered membranes, we turned to our previous finding that the ATG2A-WIPI4 complex but not ATG2A alone can tether a PI3P-containing LUV to a PI3P-free LUV (*Figure 1D*). The lipid transfer assay was adapted for this condition by replacing liposomes with DO lipids-based LUVs (100 nm), with the donors containing PI3P and the acceptors containing no PI3P (*Figure 4A*). In the absence of WIPI4, ATG2A (100 nM) slightly increased NBD fluorescence (*Figure 4B*), suggesting that ATG2A can transfer lipids between nontethered donors and acceptors but does so inefficiently. Strikingly, the addition of WIPI4 to ATG2A resulted in the acceleration of lipid transfer in a WIPI4 concentration-dependent manner, whereas the signal was not altered with WIPI4 alone (100 nM), demonstrating that WIPI4 facilitates ATG2A-mediated lipid transfer. Notably, the lipid transfer rate increased linearly with the concentration of WIPI4 and nearly saturated at the 1:1 stoichiometric concentration of WIPI4 to ATG2A (*Figure 4C*), consistent with the strong interaction between WIPI4 and ATG2A. Based on these data, the above finding that ATG2A alone can transfer lipids between tethered SUVs, and the observation that the

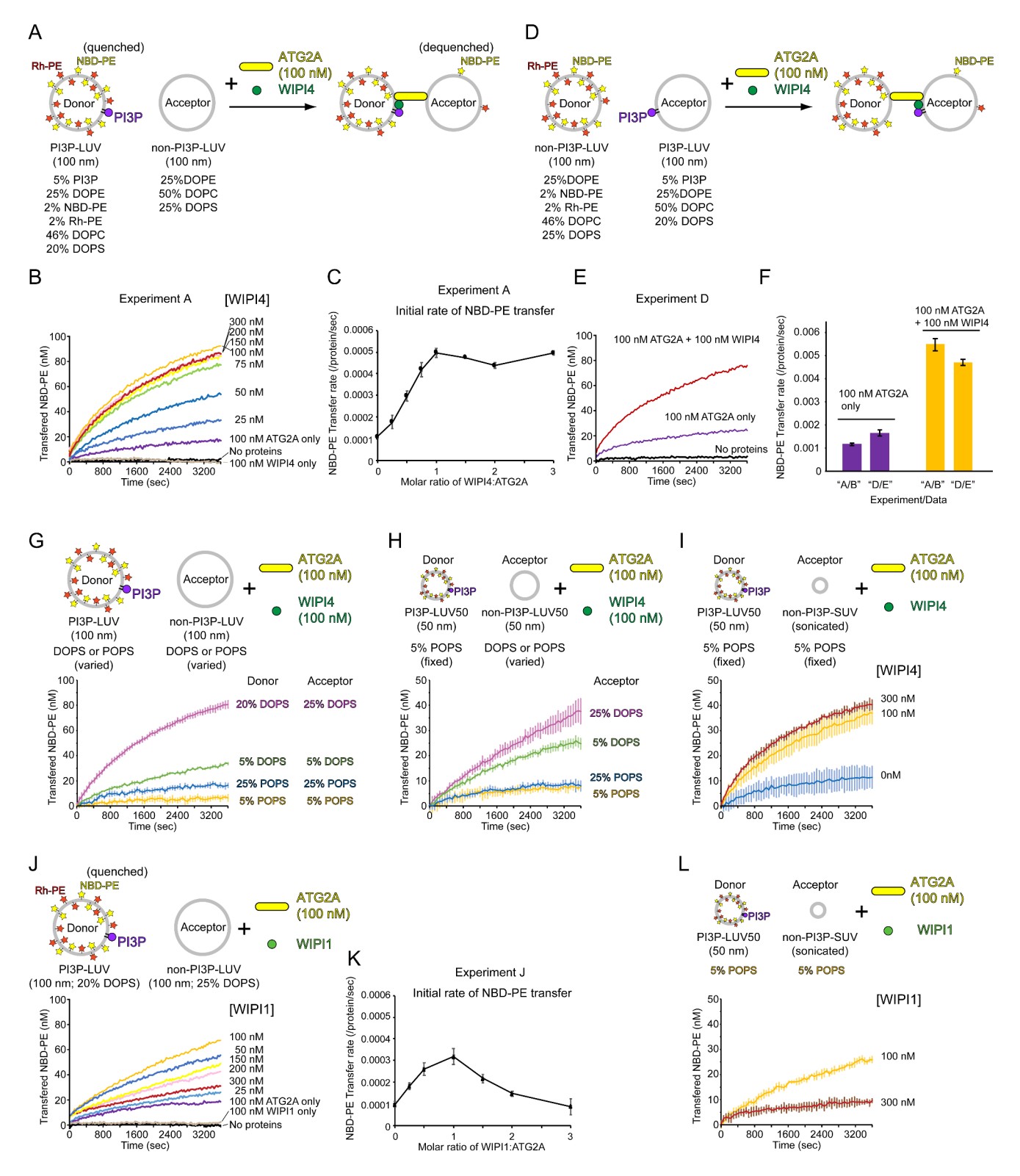

**Figure 4.** WIPI4 and WIPI1 facilitate ATG2A-mediated lipid transfer. (**A**) Diagram of the lipid transfer assay with PI3P-containing donor LUVs (extruded through a 100 nm filter, the lipid compositions; 5% PI3P, 46% DOPC, 25% DOPE, 20% DOPS, 2% NBD-PE, and 2% Rh-PE) and PI3P-free acceptor LUVs (extruded through a 100 nm filter, the lipid composition; 50% DOPC, 25% DOPE, and 25% DOPS). (**B**) Titration of WIPI4 into 100 nM ATG2A. The concentrations of WIPI4 are indicated. A representative data set is shown. (**C**) Plot of the initial NBD-PE transfer rates versus the molar ratios of WIPI4:

*Figure 4 continued on next page*

*Figure 4 continued*

ATG2A. The SD of each time point was obtained from three repeats. (D, E) Diagram of (D) and data from (E) the lipid transfer assay with 25 µM PI3P-free donor LUVs (extruded through a 100 nm filter, 46% DOPC, 25% DOPE, 25% DOPS, 2% NBD-PE, and 2% Rh-PE) and PI3P-containing acceptor LUVs (extruded through a 100 nm filter, 5% PI3P, 50% DOPC, 25% DOPE, and 20% DOPS). Note that the PI3P is in the acceptor. (F) Comparisons of the initial NBD-PE transfer rates between the experiments performed in (A, B) and (D, E). Data obtained with 100 nM ATG2A in the presence or absence of 100 nM WIPI4 are compared. (G) Lipid transfer assays with 100 nm filter-extruded LUVs with different lipid compositions. Both WIPI4 and ATG2A concentrations were fixed at 100 nM. All donor LUVs contained 5% PI3P, 2% NBD-PE, 2% Rh-PE, and 25% DOPE or POPE, for DO or PO lipids-based LUVs, respectively. The rest of the PS and PC compositions in each liposome was 20% DOPS/46% DOPC, 5% DOPS/61% DOPC, 25% POPS/41% POPC, and 5% POPS/61% POPC. All acceptors contained 25% DOPE or POPE, and the PS and PC compositions in each liposome was 25% DOPS/50% DOPC, 5% DOPS/70% DOPC, 25% POPS/50% POPC, or 5% POPS/70% POPC. Each donor was paired with the acceptor that contained the same or similar percentage of PS, as indicated. (H) Lipid transfer assays with various LUV50s. Both donor and acceptor LUV50s were prepared by extrusion through a 50 nm filter. The donor LUV50 was composed of 5% PI3P, 2% NBD-PE, 2% Rh-PE, 25% POPE, 5% POPS, and 61% POPC. This donor was used in all four experiments with different acceptor LUV50s. The lipid composition of each acceptor was identical to that of the acceptor with the same name (based on DOPS or POPS percentages) in (G). 100 nM each of WIPI4 and ATG2A were used. (I) Lipid transfer assays with 50 nm filter-extruded donors (LUV50s) and sonicated acceptors (SUVs). The same donor as in (H) (5% POPS/5% PI3P-containing LUV50) was used, and the composition of the acceptor SUV was 5% POPS, 25% POPE, and 70% POPC (the same as that of 5% POPS acceptor LUV50 in (H)). The concentration of ATG2A was 100 nM, and those of the WIPI4 used are indicated. (J) Titration of WIPI1 into 100 nM ATG2A. The experiment was performed as in (A). (K) Initial NBD-PE transfer rates in the experiment shown in (J). (L) The same experiment as (I) except that WIPI1 was tested. All experiments were performed at 25 ˚C.

DOI: https://doi.org/10.7554/eLife.45777.009

The following figure supplements are available for figure 4:

**Figure supplement 1.** Homotypic tethering of PI3P-containing donor liposomes by the WIPI4-ATG2A complex.

DOI: https://doi.org/10.7554/eLife.45777.010

**Figure supplement 2.** Homotypic tethering of PI3P-containing donor LUV50 by the WIPI4-ATG2A complex.

DOI: https://doi.org/10.7554/eLife.45777.011

**Figure supplement 3.** Cryo-EM visualization of homotypic tethering of 5% POPS-containing LUV50 by the WIPI4-ATG2A complex.

DOI: https://doi.org/10.7554/eLife.45777.012

**Figure supplement 4.** WIPI1 clusters PI3P-containing LUVs.

DOI: https://doi.org/10.7554/eLife.45777.013

ATG2A-WIPI4 complex tethers PI3P-containing LUVs to PI3P-free LUVs (*Chowdhury et al., 2018*), we concluded that lipid transfer is facilitated by ATG2A-mediated membrane tethering.

In the experiment described above, PI3P was included in the donor vesicles, orienting the ATG2A-WIPI4 complex such that the end containing WIPI4 and the CAD tip is bound to the donor and the N tip to the acceptor (*Figure 4A*). This directionality of the tethering, together with the fact that NBD fluorescence increases mostly (not exclusively) upon the transfer of fluorescent lipids from the donor to the acceptor vesicles, raises the possibility that ATG2A could be unidirectionally transferring lipids from the WIPI4 and CAD tip-bound PI3P-containing membrane to the N tip-bound PI3P-free membrane. To examine this possibility, we reversed the orientation of the ATG2A-WIPI4 complex with respect to the donor and the acceptor by including PI3P only in the acceptor vesicles (*Figure 4D*). In this reversed orientation, the signal would increase mostly upon transfer of the fluorescence lipids from the N tip-bound PI3P-free membrane to the WIPI4 and CAD tip-bound PI3P-containing membrane. Thus, a different transfer rate between the two experiments (*Figure 4A and D*) could suggest the directionality of the lipid transfer. As shown in *Figure 4E and F*, the transfer rates with this new pair of LUVs in the absence of WIPI or the presence of the stoichiometric concentration of WIPI4 were similar to those with the original LUV pair shown in *Figure 4B and C*. Thus, ATG2A-mediated lipid transfer is a bidirectional reaction, and it is therefore likely that in both experiments, non-fluorescent lipids are also being transferred in both directions.

Since the lipid composition affected lipid transfer with SUVs as described above, we also investigated the effects of the lipid composition on WIPI4-stimulated lipid transfer and found that the effects were significant. First, we reduced the fraction of DOPS in the donor and acceptor LUVs from 20% and 25%, respectively, to 5%. As a result, the lipid transfer rate decreased to ~1/2.5 of that with the high DOPS LUVs (*Figure 4G*). Lipid transfer was even slower with the LUV pairs consisting of PO lipids including 25% or 5% POPS (in both donors and acceptors), and both pairs allowed little transfer (*Figure 4G*). We examined whether these LUVs are tethered by WIPI4 and ATG2A by measuring the DLS of the PI3P-containing donor LUVs. The results confirmed that all of these LUVs were tethered by the WIPI4-ATG2A complex, but the degree of clustering decreased in the order of 20%

DOPS >5% DOPS~25% POPS>5% POPS in LUVs (*Figure 4—figure supplement 1*). Although these DLS experiments only report the degree of homotypic tethering (of the donor), and lipid transfer must occur between a donor and an acceptor, these data on homotypic tethering are indicative of the capacity of heterotypic tethering, as the donor and acceptor in each pair are composed of similar lipid compositions (except PI3P in the donor) and the N tip of ATG2A does not require PI3P in its substrate membrane for binding. Thus, the reduced membrane tethering capacities with PO lipids-based LUVs could account for the poor lipid transfer activities with these liposomes, at least in part. Perhaps the association of the N tip of ATG2A with these liposomes is not particularly stable, reducing the tethering capacity, as the other end of the complex is more likely to bind to these liposomes through the WIPI4-PI3P interaction. However, it is difficult to attribute such low lipid transfer activities solely to the reduced tethering, and it would be more reasonable to interpret the data as indicating that lipid transfer itself is less efficient with PO lipids-based membranes than DO membranes.

These results indicate that the LUV system with a more physiological lipid composition (5% POPS) cannot be employed for demonstrating WIPI4-dependent lipid transfer. To overcome this issue, we attempted to improve lipid transfer by introducing more lipid-packing defects. To do this, we prepared smaller 5% POPS-containing donor and acceptor LUVs by extrusion through a 50 nm filter (hereafter referred to as LUV50s). However, these smaller LUV50s did not improve lipid transfer (*Figure 4H*), even though DLS and cryo-EM data suggested that the PI3P-containing LUV50s can be tethered by the WIPI4-ATG2A complex (*Figure 4—figure supplements 2* and *3*). We tested different acceptor LUV50s while keeping the same donor, and the results were consistent with the data on 100 nm LUVs described above. The DO lipids-based acceptor LUV50s containing 25 or 5% DOPS improved the transfer while the PO lipids-based LUVs including 25% POPS hardly did so (*Figure 4H*). We then replaced the acceptor with SUVs containing 5% POPS. The lipid transfer between this acceptor SUV and the same donor LUV50 was inefficient in the absence of WIPI4, but the addition of WIPI4 to the 1:1 stoichiometric concentration of WIPI4 (100 nM) to ATG2A stimulated lipid transfer significantly (*Figure 4I*). Adding more WIPI4 to 300 nM only slightly improved the reaction. These results are consistent with those from the DO lipids-based experiments (*Figure 4B and C*) and with the strong association between WIPI4 and ATG2A and further suggest that lipid transfer was induced upon WIPI4-enabled tethering. Taken together, we conclude that membrane tethering is required for ATG2A-mediated lipid transfer between membranes with physiological lipid compositions, and lipid-packing defects of substrate membranes are crucial for lipid transfer.

## WIPI1 also facilitates ATG2A-mediated lipid transfer

Four mammalian WIPI paralogs have been reported to function in different stages of autophagosome biogenesis (*Proikas-Cezanne et al., 2015*). The tight binding between WIPI4 and ATG2A has been advantageous for the structural and biochemical studies conducted thus far, but we wanted to determine whether other WIPIs can facilitate ATG2A-mediated lipid transfer. Among the other three WIPIs, we succeeded in obtaining only the WIPI1 protein with sufficient stability and quantity for these biochemical assays. We first performed a titration of WIPI1 using the 20/25% DOPS-containing LUV pair (100 nm) and observed that ATG2A-mediated lipid transfer was accelerated in the presence of WIPI1 (*Figure 4J*). However, unlike with WIPI4, the increase in the transfer rate occurred only up to the WIPI1:ATG2A stoichiometric concentration; further additions of WIPI1 beyond this concentration resulted in decreases in the transfer rate (*Figure 4K*). Similarly, WIPI1 accelerated lipid transfer between the 5% POPS-containing donor LUV50s and acceptor SUVs at the stoichiometric concentration of WIPI1 to ATG2A but suppressed it at the 3:1 WIPI1 to ATG2A ratio (*Figure 4L*). Thus, WIPI1 can facilitate ATG2A-mediated lipid transfer, but it can do so only at a concentration below that of ATG2A.

In the experiments with WIPI1, we observed that the solutions containing 300 nM WIPI1 and PI3P-containing LUVs became slightly cloudy, which induced us to ask whether WIPI1 clusters those LUVs. To investigate this possibility, we performed turbidity assays with PI3P-containing LUVs and WIPI proteins and observed that the turbidity increased in a WIPI1 concentration-dependent manner (*Figure 4—figure supplement 4A and B*), indicating vesicle clustering by WIPI1. The electron micrographs of the PI3P-containing LUVs with 300 nM WIPI1 show massive liposome aggregation, confirming the clustering (*Figure 4—figure supplement 4C*). The liposome clustering was limited to WIPI1 because, in the presence of WIPI4, the turbidity did not increase (*Figure 4—figure supplement 4A and B*), and the liposomes were spread out in the EM images (*Figure 4—figure*

*supplement 4C*). The concentration dependence of WIPI1 clustering activity correlated with that of WIPI1-induced suppression of the lipid transfer activity shown above (*Figure 4J–4L*). Thus, WIPI1-induced aggregation of PI3P-containing donor LUVs would result in the sequestration of the proteins and donor LUVs from the PI3P-free acceptor LUVs, leading to the suppression of ATG2A-mediated lipid transfer at higher concentrations of WIPI1.

## WIPI4/WIPI1 enhance the association of ATG2A with PI3P-containing membranes

The facilitation of ATG2A-mediated lipid transfer by the WIPI4 and WIPI1 proteins demonstrated above suggests that ATG2A is recruited to PI3P-containing membranes by these WIPI proteins. Although we previously did not examine the membrane binding of ATG2A in the presence of WIPI4, our observation that the stable ATG2A-WIPI4 complex mediated membrane tethering involving PI3P-containing LUVs implied that WIPI4 recruits ATG2A to the membrane (*Chowdhury et al., 2018*). Because the ability of WIPI1 to recruit ATG2A has not been reported, we investigated the membrane recruitment of ATG2A by these WIPIs. We first tested the bimolecular interaction between WIPI1 and ATG2A using an affinity pull-down assay, which detected no binding of free WIPI1 to bead-immobilized ATG2A (*Figure 5A*). The same assay with WIPI4 showed its stable association with ATG2A, contrasting to the result with WIPI1. To explain the ability of WIPI1 to facilitate ATG2A-mediated lipid transfer, we hypothesized that WIPI1 and ATG2A cooperatively associate with PI3P-containing membranes, as shown previously for yeast Atg18 and Atg2 (*Kotani et al., 2018*). To test this hypothesis, we performed liposome flotation assays with 5% PI3P-containing DO LUVs (100 nm) and the WIPI4/WIPI1 and ATG2A proteins. In the experiments with WIPI4 only, the top fractions contained WIPI4, but the amount of WIPI4 was much lower than that in the bottom fractions (*Figure 5B*; lanes 4–9). In contrast, WIPI1 was mostly recovered from the top fractions (*Figure 5B*; lanes 13–18). Control experiments with LUVs without PI3P showed that both proteins remained in the bottom fractions (*Figure 5B*: lanes 19–24). Thus, both WIPIs bind to membranes in a PI3P-dependent manner, but WIPI4 binds less strongly than WIPI1, consistent with the results of the previous study that demonstrated WIPI-membrane interactions by sedimentation assays (*Baskaran et al., 2012*). As expected, ATG2A alone did not bind stably to these LUVs (*Figure 5C*; lanes 4–6). In stark contrast, in the presence of WIPI4, ATG2A was recovered almost completely from the top fractions (*Figure 5C*; lanes 7–12), and WIPI4 recovery from the top fraction (~100% at 100 nM) was improved greatly compared to that in the WIPI4-only experiments (~20% at 100 nM) (*Figure 5B*, lanes 4–9), indicating that WIPI4 and ATG2A cooperatively associate with PI3P-containing membranes. The experiments with WIPI1 yielded similar results (*Figure 5C*; lanes 13–18), demonstrating ATG2A recruitment to PI3P-containing membranes in the presence of WIPI1. Because we could not detect binding between WIPI1 and ATG2A in the affinity pull-down experiment, we do not know the mechanism for WIPI1-enhanced membrane binding of ATG2A. However, it is not unreasonable to presume that WIPI1 interacts with ATG2A but only very weakly in the absence of membranes. Even an affinity in the mM range could allow WIPI1 to keep ATG2A on a PI3P-containing membrane, as WIPI1 binds strongly to this membrane. The pull-down experiment included washing steps, which would disturb the equilibrium and remove weak binders with a fast off rate. Other methods to detect such weak interactions would require substantially more materials and higher concentrations, which are technically difficult to achieve with ATG2A. Alternatively, WIPI1 actually does not interact with ATG2A at all and only modifies the property of the membrane surface to recruit ATG2A indirectly. In either case, the WIPI1-promoted membrane recruitment of ATG2A demonstrated here suggests that WIPI1 facilitated lipid transfer by enabling membrane tethering, the same mechanism as with WIPI4. The percentages of ATG2A proteins recovered from the top fractions in the presence of WIPI1 (~60%) (*Figure 5C*; lanes 13 and 16) were slightly lower than those in the presence of WIPI4 (~100%) (*Figure 5C*; lanes 7 and 10), at least partially explaining the slower lipid transfer observed with WIPI1 than with WIPI4 at the stoichiometric concentration (100 nM) with ATG2A. In conclusion, in the presence of either WIPI protein, ATG2A binds stably to PI3P-containing membranes, providing the basis for WIPI-enabled membrane tethering and lipid transfer as demonstrated above.

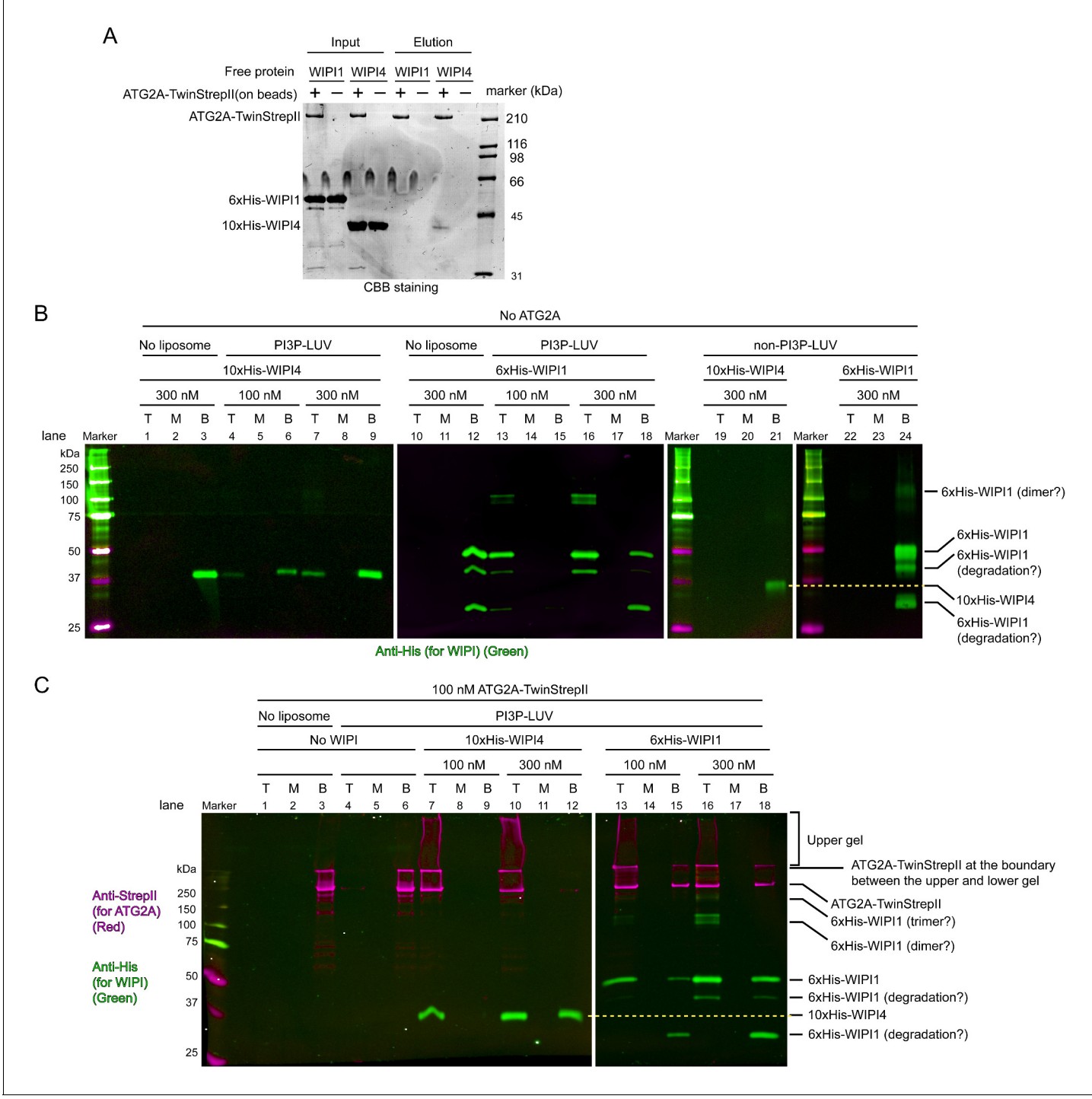

**Figure 5.** WIPI4 and WIPI1 recruit ATG2A to PI3P-containing LUVs. (**A**) Affinity pull-down experiments with ATG2A and WIPI1/WIPI4. Strep-Tactin Superflow affinity beads loaded with or without ATG2A-TwinStrepII were mixed with free WIPI1/WIPI4 proteins. After wash, proteins were eluted. Inputs and eluted fractions were analyzed by SDS-PAGE with CBB staining. (**B**) Liposome flotation assays to detect membrane binding of WIPI4 and WIPI1. DO lipids-based LUVs (5% PI3P, 49% DOPC, 25% DOPE, 20% DOPS, and 1% DiD) were used at 25 μM. The control LUVs were prepared without PI3P. Protein concentrations are indicated. The top (T), middle (M), and bottom (B) fractions were collected after centrifugation and analyzed by western blotting. Bands were detected using an anti-His antibody on the 800 nm channel of a LI-COR infrared fluorescence scanner. (**C**) Liposome flotation assays to examine ATG2A (100 nM) recruitment to PI3P-containing membranes by WIPI proteins. The experiments were performed as in (**B**). ATG2A-TwinStrepII was detected using an anti-StrepII antibody on the 700 nm channel of the infrared scanner (magenta). His-WIPI proteins were detected as in (**B**) on the 800 nm channel (green). 10% acrylamide SDS-PAGE gels were used in all experiments.

DOI: https://doi.org/10.7554/eLife.45777.014

## Discussion

Recent yeast studies suggested that the Atg2-Atg18 complex directly tethers the phagophore to the ER (*Gómez-Sánchez et al., 2018*; *Kotani et al., 2018*). Atg18 is on the edge of PI3P-positive phagophores (*Graef et al., 2013*; *Suzuki et al., 2013*), directing the CAD tip of Atg2 to the phagophore and the N tip to the ER (*Graef, 2018*). ER anchoring by the N tip was demonstrated by Kotani et al., who created an autophagy-deficient Atg2 mutant by deleting the N-terminal 21 residues and restored autophagic activity by genetically fusing the transmembrane domain of the ER resident Sec71 protein to the N-terminus of the mutant (*Kotani et al., 2018*). In mammals, the precise location of ATG2A/B on the expanding phagophore had not been established until very recently, but during the revision of this manuscript, Valverde et al. reported that ATG2A is localized to the ER-phagophore contact site together with WIPI2 (*Valverde et al., 2019*). The edge of the phagophore has been suggested to form a PI3P-enriched transient region called the 'omegasome' (*Axe et al., 2008*). The mechanism by which ATG2A localizes to the omegasome is not known but likely involves its interaction with WIPI proteins including WIPI4, which has been shown to localize to the omegasome (*Lu et al., 2011*). Thus, we place the ATG2-WIPI complex between the ER and the phagophore edge, similar to the location suggested for the yeast Atg2-Atg18 complex (*Kotani et al., 2018*). Direct tethering of the ER and the phagophore edge by the ATG2-WIPI complex would allow the transfer of ER lipids to the phagophore, driving phagophore expansion (*Figure 6A*). This model of phagophore expansion based on non-vesicular lipid transfer explains why and how the edge of the phagophore remains associated with the ER during its expansion (*Graef et al., 2013*; *Suzuki et al., 2013*) and why autophagosome membranes are of thinner types like the ER membrane (*Arstila and Trump, 1968*) but are devoid of ER membrane-bound proteins/markers such as cytochrome P-450, Dpm1, and GFP-HDEL (*Suzuki et al., 2013*; *Yamamoto et al., 1990*). Moreover, the recent observation that the ER-staining lipophilic dye octadecyl rhodamine B also stains the phagophore in yeast (*Hirata et al., 2017*) can be explained by the ATG2-mediated transfer of this lipophilic dye from the ER to the phagophore.

The rate of ATG2-mediated lipid transfer depends on the lipid-packing defects of the substrate membranes and negatively charged lipids. On the one hand, the phagophore edge is likely abundant in both of these factors: the lipid packing defects created by its high curvature with a diameter of ~30 nm (*Nguyen et al., 2017*) and the negative charges resulting from the local production of PI3P. On the other hand, these parameters of the ATG2-associating site of the ER membrane are much less clear. In yeast, the phagophore edge has been shown to be located at the ER exit site (ERES) (*Graef et al., 2013*; *Suzuki et al., 2013*), the place where COPII vesicles are generated. In mammals, in addition to the ERES, it has also been suggested that the ERGIC and the ER-mitochondria contact site are adjacent to the phagophore (*Ge et al., 2013*; *Ge et al., 2017*; *Hamasaki et al.,*

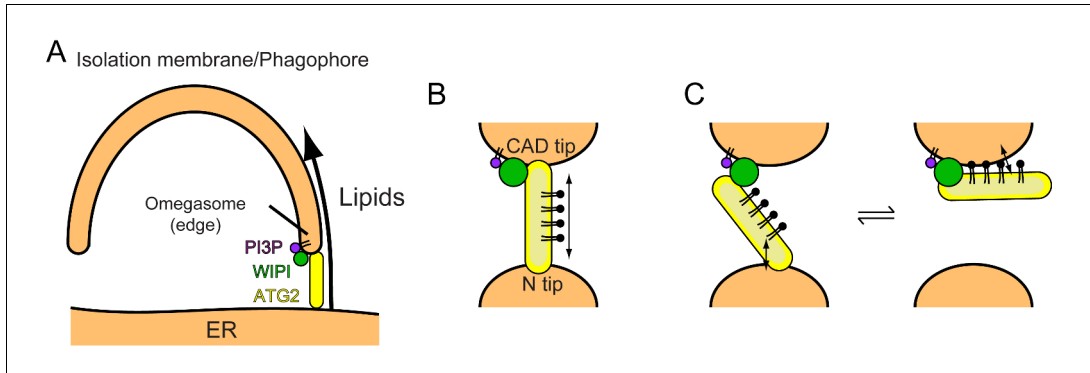

**Figure 6.** Models of phagophore expansion and ATG2-mediated lipid transport. (**A**) Illustration of ATG2-mediated transport of lipids from the ER to the phagophore. (**B**) The bridge model for WIPI-ATG2-mediated lipid transfer between a highly curved PI3-containing membrane and a non-PI3P-containing membrane. (**C**) The ferry model. ATG2 dynamically associates with both membranes while stably anchored to the PI3-containing membrane through the flexibly bound WIPI.
DOI: https://doi.org/10.7554/eLife.45777.015

*2013*; *Hayashi-Nishino et al., 2009*; *Karanasios et al., 2016*). All of these particular regions of the ER are highly dynamic and could contain tubular membranes, implying that the N-terminus of ATG2 could be bound to a highly curved location within these ER regions. In terms of the negative charges of those membranes, PI has been shown to be enriched in the phagophore-contacting ER subdomain (*Nishimura et al., 2017*). PI synthase localizes to this ER subdomain and generates PI molecules, which could be transferred to the phagophore by ATG2 to serve as the precursor of PI3P. Therefore, it is possible that the phagophore-contacting ER membrane is also promotive of ATG2-mediated lipid transfer.

One of the ultimate goals of this work is to fully explain how ATG2A-mediated lipid transfer delivers a sufficient number of lipids to create an autophagosome, which can be composed of tens of millions of lipids (~25 million for a double-membrane vesicle with a diameter of 1 μm). Since the exact number of ATG2 molecules at the ER-phagophore contact site is not known, we cannot accurately calculate the theoretically achievable bulk rate of ATG2-mediated lipid transport for phagophore expansion. However, the rate of lipid transfer by individual ATG2A molecules estimated above seems to be too slow to accomplish this enormous task and therefore must be accelerated much more. This issue is closely related to another important one: in order to drive phagophore expansion, the equilibrium of ATG2-mediated lipid transfer must be shifted toward the phagophore. Currently, no mechanism that enables such fast and unidirectional lipid transport for phagophore expansion is known. In a thermodynamic point of view, unidirectional lipid transport must consume energy, and by consuming energy, lipid transport could be greatly accelerated. The type of energy utilized for this process is an important unknown. We speculate that some yet-unidentified proteins are responsible for this energy-expending task. When these proteins assist the lipid transport by ATG2, the two lipid transfer-promoting factors, lipid-packing defects and negative charges of substrate membranes, could be less crucial.

The demonstration that WIPI proteins can facilitate ATG2A-mediated lipid transfer suggests that PI3P dynamics control phagophore expansion: the number of PI3P molecules on the phagophore would determine the number of ATG2 proteins, thereby regulating the overall rate of phagophore growth. A reduction in the PI3P level, which occurs toward the end of autophagosome formation (*Axe et al., 2008*), would release ATG2 from the point of ER-phagophore contact, leading to the disruption of the contact and the cessation of phagophore expansion. This model of the regulation of phagophore expansion is supported by the previous observation that the overexpression of the PI3P-phosphatase myotubularin-related phosphatase three generates significantly smaller autophagosomes (*Taguchi-Atarashi et al., 2010*). According to this model, the PI3P effectors Atg18/WIPIs play a pivotal role in correlating PI3P dynamics to phagophore expansion. Atg18 partners primarily with Atg2, suggesting its linear relationship with phagophore expansion. In contrast, the direct link between WIPIs and ATG2A/B has been limited to WIPI4. Notably, our work now adds WIPI1 to this link, raising the possibility that the yet-untested WIPI2 and WIPI3, which, like WIPI1, do not interact strongly with ATG2A/B (*Bakula et al., 2017*), might also facilitate ATG2A recruitment. The high sequence identity (~80%) between WIPI1 and WIPI2 (*Polson et al., 2010*) supports this potential role of WIPI2.

The WIPI1-induced clustering of PI3P-containing liposomes was likely induced by the self-oligomerization of WIPI1, as WIPI1 oligomer-like bands were detected in SDS-PAGE (*Figure 5C*), and oligomerization has been observed with other Atg18/WIPI proteins on membranes and in solution (*Baskaran et al., 2012*; *Gopaldass et al., 2017*; *Scacioc et al., 2017*). It is noteworthy that the self-oligomerization of Atg18 on giant unilamellar vesicles tubulates the membranes (*Scacioc et al., 2017*), as such membrane tubulation activity may lead to the formation of the omegasome, which has been visualized by EM as a PI3P-positive thin tubular membrane at the phagophore edge (*Uemura et al., 2014*). Under our experimental condition, we did not observe tubular membranes with WIPI1, but it is tempting to speculate that with help from other factors, WIPI proteins also might induce membrane tubulation.

Our results showing membrane tethering-facilitated lipid transfer activity of ATG2A suggest a 'bridge' model, in which ATG2 stably tethers two membranes and transfers lipids between them (*Figure 6B*). Lipids are loaded from one membrane onto one of the tips of ATG2A, then slide through the extended hydrophobic cavity to reach the other tip, and are finally unloaded onto the other membrane. A similar model has been proposed by Osawa et al., during our revision, who demonstrated that yeast Atg2 transfers lipids between SUVs but poorly between LUVs

(*Osawa et al., 2019*). However, Valverde et al., who also reported the lipid transfer capacity of ATG2A, demonstrated that a short (345 residues) ATG2A N-terminal fragment can not only mediate lipid transfer in vitro but also rescue autophagy in ATG2A/B double knockout cells (under overexpressed conditions) (*Valverde et al., 2019*). It should be noted that in their in vitro experiments, which lacked WIPI proteins, they tethered LUVs artificially to promote lipid transfer by both the full-length and the N-terminal fragment of ATG2A. While joining our conclusions with those of Osawa et al. to underscore the importance of membrane tethering for efficient lipid transfer, their data suggest that lipid transfer could be achieved by the shuttling of ATG2A between membranes. At least the N-terminal fragment, which lacks the other end (CAD tip) of ATG2A and therefore should be incapable of membrane tethering, is most likely operating by shuttling. Such a dynamic 'ferry' model challenges the static 'bridge' model. In our previous negative stain EM study, we observed ATG2A molecules bound to not only two SUVs (tethered) but also one SUV (untethered) (*Chowdhury et al., 2018*), which could indicate a dynamic aspect of the ATG2A-membrane interaction. In principle, membranes could be tethered by dynamically associating molecules if multiple molecules are involved in the tethering (a multivalent effect). WIPI4 is bound to a short flexible region of ATG2A (*Chowdhury et al., 2018*), which may allow ATG2 to undergo a swing-like movement and thereby dynamically associate with both membranes without dissociating completely from the PI3P-containing membrane (*Figure 6C*). These intriguing models are difficult to sort out by experiments based on bulk solutions but collectively provide a foundation for future biophysical studies to dissect the underlying mechanisms, including those of lipid loading and unloading. On the structural side, Osawa et al. succeeded in visualizing a lipid molecule (PE) in the hydrophobic cavity of the Atg2 N-terminus by X-ray crystallography, providing a structural basis of lipid binding of the N-terminus. Further information regarding the extended cavity as well as the other end of the molecule (CAD tip) will require another breakthrough in structural studies on longer constructs.

## Materials and methods

### Key resources table

| Reagent type (species) or resource | Designation | Source or reference | Identifiers | Additional information |
|---|---|---|---|---|
| Recombinant DNA reagent | pAceBac1 GST-human ATG2A-TwinStrepII | *Chowdhury et al., 2018* | | Plasmid DNA |
| Recombinant DNA reagent | pAceBac1 10xHis-human WIPI4 | *Chowdhury et al., 2018* | | Plasmid DNA |
| Recombinant DNA reagent | pFASTBAC 6xHis-human WIPI1 | this paper | | Plasmid DNA; Human WIPI1(1–446) cloned into a modified pFASTBAC-HTa (Invitrogen) |
| Antibody | Anti-His tag, mouse monoclonal | Millipore | Cat. #05–949 | WB (1:500) for 10xHis-WIPI4, WB (1:2000) for 6xHis-WIPI1 |
| Antibody | Anti-NWSHPQFEK (StrepII), rabbit polyclonal | GenScript | Cat. #A00626 | WB (1:1000) |
| Antibody | Anti-Mouse IRDye 800CW, goat polyclonal | LI-COR Biosciences | 827–08364 | WB (1:20000) |
| Antibody | Anti-Rabbit IRDye 680, goat polyclonal | LI-COR Biosciences | 827–08367 | WB (1:20000) |
| Chemical compound, drug | Sodium dithionite | Sigma-Aldrich | Cat. #157953 | |
| Chemical compound, drug | PI3P | Avanti Polar Lipids, Inc | Cat. #850150P | |
| Chemical compound, drug | DOPC | Avanti Polar Lipids, Inc | Cat. #850375C | |

*Continued on next page*

*Continued*

| Reagent type (species) or resource | Designation | Source or reference | Identifiers | Additional information |
|---|---|---|---|---|
| Chemical compound, drug | DOPE | Avanti Polar Lipids, Inc | Cat. #850725C | |
| Chemical compound, drug | DOPS | Avanti Polar Lipids, Inc | Cat. #840035C | |
| Chemical compound, drug | POPC | Avanti Polar Lipids, Inc | Cat. #850457C | |
| Chemical compound, drug | POPE | Avanti Polar Lipids, Inc | Cat. #850757C | |
| Chemical compound, drug | POPS | Avanti Polar Lipids, Inc | Cat. #850757C | |
| Chemical compound, drug | Bovine liver (bl) PI | Avanti Polar Lipids, Inc | Cat. #840042C | |
| Chemical compound, drug | NBD-PE 18:1 | Avanti Polar Lipids, Inc | Cat. #850757C | |
| Chemical compound, drug | Rh-PE 18:1 | Avanti Polar Lipids, Inc | Cat. #810150C | |
| Chemical compound, drug | DiD (1,1'-Dioctadecyl-3,3,3',3'-tetramethylindodi carbocyanine perchlorate) | Marker Gene Technologies | Cat. #M1269 | |

## Protein expression and preparation

Human ATG2A and WIPI4 were expressed in *Spodoptera frugiperda* 9 (Sf9) insect cells and purified as described previously (*Chowdhury et al., 2018*). Human WIPI1 was expressed and purified using the same procedure used for WIPI4. In brief, ATG2A was expressed in Sf9 insect cells as a fusion to an N-terminal TEV protease-cleavable glutathione S-transferase (GST) and to a C-terminal PreScission protease-cleavable TwinStrepII (TS) tag. The expressed proteins were purified on glutathione sepharose beads (GoldBio) and eluted upon on-beads TEV cleavage in the presence of TritonX-100. The eluted protein (ATG2A-TS) was further purified on StrepTactin XT beads (IBA). Beads were washed extensively in wash buffer 20 mM HEPES (pH 7.5), 150 mM NaCl, 1 mM Tris(2-carboxyethyl) phosphine (TCEP) to remove TritonX-100 and eluted in the same buffer supplemented with 50 mM biotin. Purified ATG2A-TS was dialyzed against wash buffer to remove biotin. WIPI4 and WIPI1 were expressed in Sf9 cells with N-terminal TEV protease-cleavable ten-histidine (10 × His) and six-histidine (6 × His) tags, respectively. Both proteins were purified by Ni affinity chromatography, anion exchange chromatography, and size exclusion chromatography. WIPI4 and WIPI1 proteins from which the His tags were removed were used in the lipid transfer assays, and those with uncleaved His tags (10 × His-WIPI4 and 6 × His-WIPI1) were used in the liposome flotation assays.

## NBD-PE lipid extraction assay

Lipids (Avanti Polar Lipids) were mixed in a glass tube at a molar ratio of 80% 1,2-dioleoyl-*sn*-glycero-3-phosphocholine (DOPC) and 20% 1,2-dioleoyl-sn-glycero-3-phosphoethanolamine-N-(7-nitro-2–1,3-benzoxadiazol-4-yl) (NBD-PE) dissolved in chloroform and dried under a stream of nitrogen gas. The obtained lipid film was further dried under a vacuum for 30 min and then hydrated in 20 mM HEPES (pH 7.5) and 150 mM NaCl. The hydrated lipids were subjected to seven freeze-thaw cycles using liquid nitrogen and a water bath set to 42 ℃. The resulting LUV solution was passed more than 20 times through a polycarbonate filter membrane with a pore size of 100 nm using the extruder from Avanti Polar Lipids, Inc. The prepared LUVs were stored at 4 ℃ in the dark. A 200 nM concentration of ATG2A protein was mixed with 100 nM LUVs in 20 mM HEPES (pH 7.5), 150 mM NaCl, and 1 mM TCEP (total volume, 37.5 µL). After incubation for 1 hr at ~22 ℃, an equal volume of 80% Nycodenz (Accurate Chemical) solution was added to the protein-LUV solution and mixed thoroughly. Then, the mixture was placed at the bottom of a centrifuge tube. A total of 475 µL of

30% Nycodenz solution was placed above the bottom layer of solution, and 50 µL of buffer with no Nycodenz (20 mM Hepes (pH 7.5), 150 mM NaCl, and 1 mM TCEP) was then placed on the top. The tubes were centrifuged in an SW55Ti rotor (Beckman Coulter) at 279,982 × g (max) for 1.5 hr at 18 °C. After centrifugation, the top 520 µL was removed, and the bottom fraction (80 µL) was then collected and subjected to SDS-PAGE. The gels were scanned on a Typhoon 9410 imager (GE Healthcare) to quantify NBD fluorescence, followed by CBB staining and scanning on an Odyssey imager (LI-COR Biosciences) for protein visualization and quantification of the protein concentration. The bottom fractions were also subjected to native PAGE. Decyl maltoside was added to the sample fractions at a final concentration of 0.2% to prevent aggregation of the ATG2A protein in the wells of the gel. The gel was scanned on the Typhoon 9410 imager to detect NBD fluorescence, followed by CBB staining to detect proteins.

## NBD-PE lipid unloading assay

LUVs containing 100% DOPC were prepared exactly as described above. ATG2A loaded with NBD-PE was collected from the bottom fraction of the extraction assay described above, and DOPC-containing LUVs were added to this fraction at a final concentration of 100 nM. After a 1 hr incubation, 80% Nycodenz solution was added to the mixture to adjust the Nycodenz concentration to 40%. This solution (75 µL) was placed at the bottom of a centrifuge tube, and 30% Nycodenz (475 µL) and 0% Nycodenz (50 µL) were sequentially placed above the bottom solution. The tube was centrifuged as described above. The top and bottom fractions (80 µL each) were collected and subjected to SDS-PAGE analysis. The gels were scanned on Typhoon 9410 imager for quantification of fluorescence, followed by CBB staining to visualize proteins.

## Lipid transfer assay

The liposomes were prepared with lipid compositions as described in figure legends. All donor liposomes contained 2% NBD-PE and 2% Rh-PE. All liposomes contained 25% DOPE or 25% POPE for DO or PO lipids-based liposomes, respectively. The percentage of PI3P was 5% when PI3P was included. All liposomes were prepared as described above. Proteins were mixed with 25 µM (concentration of lipids) donor and acceptor vesicles in a cuvette and placed in a fluorimeter (Photon Technology International) at 30 °C (only for the experiment shown in *Figure 3B*) or 25 °C. NBD fluorescence was recorded at an excitation wavelength of 460 nm and an emission wavelength of 535 nm every 1 s (*Figure 3*) or 30–60 s (*Figure 4*). At the end of each reaction, Triton X-100 was added to the reaction mixture to a final concentration of 0.1% (v/v) to solubilize all lipids and therefore maximize NBD fluorescence. This fluorescence intensity was set to the total NBD-PE concentration (fully transferred NBD-PE) and the fluorescence intensity of the liposome in the absence of proteins was set to zero transferred NBD-PE. For the dithionite experiments, sodium dithionite dissolved in 50 mM Tris (pH 10) was added to the postreaction mixture at a final concentration of 5 mM. Rates were obtained by linear regression of the initial points in the time course data (5–25 s for *Figure 3* and 30-300 seconds for *Figure 4*).

## Affinity pull-down assay

ATG2A-TS was loaded onto Strep-Tactin Superflow beads in buffer containing 20 mM HEPES (pH 7.5), 150 mM NaCl, 0.5 mM TCEP, 20% glycerol, and 0.1% Triton X-100. After loading, the beads were rigorously washed with buffer and were then mixed with free 10 × His-WIPI4 or 6 × His-WIPI1 proteins. After ~5 min of incubation, the tubes containing the mixtures were centrifuged to sediment the beads. The beads were washed three times, after which proteins were eluted in the same buffer supplemented with 5 mM desthiobiotin. The input and eluate solutions were subjected to SDS-PAGE.

## Liposome flotation-based binding assay

Liposome flotation assays were performed as described previously (*Chowdhury et al., 2018*). Briefly, ATG2A, WIPI4, and WIPI1 proteins were mixed with 25 µM LUVs (5% PI3P, 49% DOPC, 25% DOPE, 20% DOPS, and 1% 1,1′-Dioctadecyl-3,3,3′,3′-tetramethylindodicarbocyanine perchlorate (DiD) (Marker Gene Technologies)) as indicated in *Figure 5B and C* in 150 µL of buffer containing 20 mM HEPES (pH 7.5), 150 mM NaCl, 1 mM TCEP, and 40% Nycodenz at the bottom of centrifugation

tubes. A layer of 400 µL of 30% Nycodenz buffer was placed on the top of each sample, and a layer of 50 µL of 0% Nycodenz buffer was then placed on top. Tubes were centrifuged as described above. The top (150 µL), middle (300 µL), and bottom (150 µL) fractions were collected from the top and subjected to western blotting. ATG2A-TwinStrepII and His-WIPI proteins (10 × His-WIPI4 and 6 × His-WIPI1) were probed with rabbit anti-StrepII (GenScript) and mouse anti-His tag (Millipore) antibodies, respectively. Goat anti-mouse IRDye 800CW and goat anti-rabbit IRDye 680 secondary antibodies (LI-COR Biosciences) were used for infrared fluorescence detection of the bands on LI-COR Odyssey scanner (LI-COR Biosciences).

## Turbidity assay

LUVs at a lipid concentration of 25 µM ([5% PI3P, 50% DOPC, 25% DOPE, and 20% DOPS] or [5% PI3P, 65% POPC, 25% POPE, and 5% POPS]) were mixed with 100, 200, or 300 nM WIPI1 or WIPI4 in 20 mM HEPES (pH 7.5), 150 mM NaCl, and 1 mM TCEP and incubated at ~22 °C in a 60 µL cuvette placed in an Ultraspec 2000 UV spectrophotometer (Pharmacia Biotech). The optical density (OD) at 400 nm was recorded every 10 s.

## Negative stain electron microscopy

A 3 µL drop of solutions containing liposomes and proteins was placed on a glow-discharged continuous carbon grid (Electron Microscopy Science) and stained with 2% uranyl acetate. Grids of the samples from the lipid extraction experiments (Figure 2—figure supplement 2) were imaged with a Morgagni transmission electron microscope (TEM) operating at 80 keV at a magnification of 57,000 ×. Grids of the uranyl acetate-stained samples containing WIPI1 and PI3P-containing liposomes were imaged with an FEI Tecnai F20 TEM operating at 200 keV at a magnification of 11,500 ×.

## Cryo-EM of the ATG2A-WIPI4 complex

A 4 µL drop of the ATG2A-WIPI4 complex (0.1 mg/ml) was applied onto a glow-discharged Quantifoil R 1.2/1.3 300 mesh grid and an excess liquid was removed by a filter paper. The grid was immediately vitrified in liquid ethane using a manual plunger in a cold room. Grids were imaged with a 200-keV Talos Arctica TEM (Thermo Fisher Scientific) equipped with a K2 Summit direct electron detector (Gatan) at a magnification of 36,000 × (1.15 Å/pixel). Data were acquired in the movie mode with an electron dose of ~65e-/Å$^2$ with a defocus range of −0.7 to −2.0 µm. A total of 780 micrographs were collected in an automated manner using the software Leginon (Suloway et al., 2005). Frame alignment were performed using MotionCor2 software (Zheng et al., 2017b) in the Appion data processing pipeline (Lander et al., 2009) during data collection. CTF parameters were estimated with gCTF (Zhang, 2016). Single-particle analysis was carried out in Relion 3.0 (Zivanov et al., 2018). Particles with an elongated shape were manually picked and subjected to 2D classification. The resulting templates were used for autopicking, yielding 70,000 particles, which were extracted with a pixel size of 4.6 Å/pixel and a box size of 64. Several rounds of 2D classification were carried out to yield the images shown in Figure 1E. Attempts to visualize secondary structure elements in 2D averages, which are necessary for high-resolution structure determination, were made but unsuccessful.

## Cryo-EM of liposomes

A 4 µL sample containing 300 µM (lipid concentration) LUV50s consisting of 5% PI3P, 65% POPC, 25% POPE, 5% POPS) with or without 100 nM each of WIPI4 and ATG2A was placed onto a grow-discharged lacey formvar/carbon-coated EM grid (PELCO TEM) and vitrified as described above. The grids were imaged with a 200-keV Talos TEM (Thermo Fisher Scientific) equipped with a Ceta CCD camera at a magnification of 73,000 ×. Data were acquired with an electron dose of ~40 e-/Å$^2$ with a defocus of −3.0 or −5.0 µm.

## DLS

DLS experiments were performed as described previously on DynaPro Plate Reader II (Wyatt Technology) (Chowdhury et al., 2018). Proteins (100 nM WIPI4, 100 nM ATG2A, or both) and liposomes with a lipid concentration of 60 µM were mixed in 50 µL buffer consisting of 20 mM HEPES pH7.5, 150 mM NaCl, and 0.5 mM TCEP. The donor liposomes prepared for the lipid transfer experiments

were used. The sample mixtures were incubated for 20 min at 25 ˚C and then subjected to a DLS measurement. The laser power and the attenuation factor were set to auto mode. The results were analyzed using the program DYNAMICS included in the instrument package.

## Acknowledgements

We thank Dr. Kazuto Ohashi for his contribution to preliminary studies for this project, Drs. Gabriel Lander and Saikat Chowdhury for initial help and training on cryo-EM data collection, and Dr. Imre Berger for providing the MultiBac expression system. We are grateful to Dr. Ashok Deniz for his generous advice and critical reading of the manuscript. This research was supported by a grant from the National Institute of General Medical Sciences (R01-GM092740). Computational analyses of EM data were performed using shared instrumentation at The Scripps Research Institute funded by NIH S10OD021634.

## Additional information

### Funding

| Funder | Grant reference number | Author |
| --- | --- | --- |
| National Institute of General Medical Sciences | R01GM092740 | Takanori Otomo |

The funders had no role in study design, data collection and interpretation, or the decision to submit the work for publication.

### Author contributions

Shintaro Maeda, Resources, Data curation, Formal analysis, Validation, Investigation, Visualization, Writing—review and editing; Chinatsu Otomo, Resources, Data curation, Formal analysis, Validation, Investigation; Takanori Otomo, Conceptualization, Resources, Data curation, Formal analysis, Supervision, Funding acquisition, Validation, Investigation, Visualization, Methodology, Writing— original draft, Project administration, Writing—review and editing

### Author ORCIDs

Takanori Otomo (iD) https://orcid.org/0000-0003-3589-238X

### Decision letter and Author response

Decision letter https://doi.org/10.7554/eLife.45777.019
Author response https://doi.org/10.7554/eLife.45777.020

## Additional files

### Supplementary files

• Transparent reporting form
DOI: https://doi.org/10.7554/eLife.45777.016

### Data availability

All data generated and analysed during this study are included in the manuscript and supporting files.

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
