## [Decision Letter]

Thank you for submitting your article "The autophagic membrane tether ATG2A transfers lipids between membranes" for consideration by *eLife*. Your article has been reviewed by three peer reviewers, one of whom served as a guest Reviewing Editor, and the evaluation has been overseen by Vivek Malhotra as the Senior Editor. The following individual involved in review of your submission has agreed to reveal their identity: Bruno Antonny (Reviewer #3).

The reviewers have discussed the reviews with one another and the Reviewing Editor has drafted this decision to help you prepare a revised submission.

Summary:

This manuscript identifies ATG2A as a lipid transfer protein in vitro and proposes that such activity is involved in autophagosome expansion during autophagy. This is a new message and an interesting working hypothesis for the mechanism of autophagosome formation. So in principle this work should be suitable for publication in *eLife*. However, all three reviewers agreed that a few additional experiments and controls would be required to strengthen some of the conclusions. All experiments listed below use the techniques already described by the authors so should not take much time.

Essential revisions:

- Test ATG2A lipid transfer using different liposome compositions as indicated by reviewer 3 (point 1).

- Include an ATG2 lipid-binding deficient mutant in the lipid transfer assays (reviewer 1, point 1).

- Perform additional controls for liposome integrity (reviewer 2 points 1 and 2) and PI3P-dependence in WIPI1 membrane binding.

*Reviewer #1:*

Maeda et al. characterize human ATG2A as a lipid transfer protein in vitro. Following up on their previous structural work, they now show that Atg2 is capable of extracting lipids from one membrane and unloading them on to a different one. Furthermore, it is shown that the lipid transfer is greatly enhanced by WIPI1 and WIPI4 under conditions in which one of the vesicles contains PI3P, a result consistent with their previous study. However, in contrast with WIPI4, WIPI1 does not appear to bind directly to ATG2A in solution and appears capable of inducing liposome clustering. Overall, the identification of ATG2A as a lipid transfer protein is a significant advance to the field. Moreover, most the data is of good quality and the experiments are presented in a clear and logical manner. However, I find that a few controls are missing.

1) In my opinion the paper would become stronger if an ATG2 lipid-binding deficient mutant was included. I am not sure whether such a mutant is available but if yes, its inclusion would make conclusions stronger.

2) The effect on PI3P on ATG2A/WIPI4 binding to membranes has been previously characterized by the authors (Chowdhuri et al., 2018). However, that is not the case for WIPI1. Thus, in face of the data presented, it is not possible to conclude that PI3P plays a critical role in WIPI1 membrane binding as its binding was not tested in the absence of PI3P.

3) In my opinion, the results on the cooperativity between WIPI1 and ATG2A are overinterpreted, in particular considering the effect of WIPI1 on liposome morphology and on the ATG2A binding to membranes with packing defects. I wonder whether by altering liposome morphology/curvature, WIPI1 generates membrane packing defects which in turn facilitate ATG2A binding. Perhaps this possibility should be considered and discussed.

*Reviewer #2:*

The manuscript by Maeda et al. reports that the human ATG2A protein is capable of transferring lipids between artificial bilayers. It was previously shown by the same lab and others that ATG2 in complex with the WIPI proteins can tether membranes (PMIDs: 30254161, 30185561, 29848619). In addition, it was shown that the VPS13 proteins, which show some similarity to the ATG2 proteins are lipid transfer proteins (PMID: 30093493). in vivo, ATG2 proteins are required for expansion of the nascent autophagosomal membrane. The results presented in this manuscript suggest that ATG2A aids autophagosome formation by transferring lipids between the tethered ER membrane and the growing autophagosomal membrane.

Even though the manuscript lacks in vivo validation, the data presented by Maeda et al. are an important contribution to the autophagy and provide a crucial biochemical activity for one of the central autophagy factors. As such it will be interesting for the readers of *eLife*. However, some additional controls should be conducted in order to corroborate the finding that ATG2 is a bona fide lipid transfer protein.

1) Figure 2: The authors should test if there are liposomes remaining in the bottom fractions. This could be done for example by electron microscopy or DLS. In addition, the authors should provide a quality control for the integrity of the liposomes containing 20% NBD-PE. Furthermore, the ATG2A input needs to be shown for 2D, I. Also, in 2B it seems there is significantly more overall fluorescence in the tube +ATG2A. Initial levels of fluorescence (input) should be shown.

2) Along similar lines, the authors should test if the addition of ATG2 or ATG2-WIPI4 does disrupt the liposomes used in Figures 2 and 4.

3) The authors show in Figure 5 that WIPI1 recruits ATG2 to the membrane but a direct interaction is not detectable (Figure 5A). The authors should show the same pulldown assay for WIPI4. Not only will it give an idea of the ATG2A/WIPI4 interaction strength (that was shown only indirectly as the authors explain in the text) but also will serve as a positive control for the ATG2A/WIPI1 pulldown. In addition, the authors should use more sensitive assays to demonstrate a direct interaction between WIPI1 and ATG2. An alternative explanation would be that WIPI1 disturbs the membrane that these defects are recognized by ATG2.

4) The data shown in Figure 6 are somewhat out of context and it is unclear how the liposome clustering activity is integrated into the process of autophagy.

5) In the Introduction, the authors refer to the crystal structure of Vps13 N-terminal part from *Chaetomium thermophilum* (Introduction, fourth paragraph) described in Kumar et. al mentioning that it can accommodate ~10 lipid molecules, where in fact the number of lipid molecules was estimated for N-terminal domain of Vps13α from *S. cerevisiae* in Mass Spec analysis. This should be clarified in the text.

6) Figure 1: Adding residues numbers to the cartoon representation of ATG2A would give a better idea of the size of particular domains. The authors should depict the borders of the crystallized homology domain on the cartoon representation of ATG2A.

7) Discussion: The discussion about the possible regulation of the lipid loading/unloading appears to be too speculative.

*Reviewer #3:*

Previous work from the Otomo lab and from other labs have shown that the autophagy protein Atg2, both in yeast and in mammals, is capable of tethering membranes and that this activity is important for the expansion of the phagophore. Here, using other biochemical assays, Maeda et al. demonstrates that Atg2A is also able to extract and deliver lipids from/to artificial membranes. The coupling between membrane tethering and lipid extraction/delivery allows Atg2 to act as an efficient lipid transfer protein. From these observations, the authors propose a new and attractive mechanism for the expansion of the phagophore: the organelle gets lipids from the ER by a lipid transfer mechanism and not by vesicular traffic. The overall message of the work is novel and gives a key working hypothesis for the mechanism of autophagosome formation. As such, it should be suitable for publication in *eLife*.

A few additional experiments and textual changes might clarify some issues.

1) My main concern is about the composition of the liposomes. In Figure 1 where the authors test the ability of Atg2 to extract and deliver lipids, they use a mixture of DOPC and NBD-PE. In contrast, in the figures where they test the ability of Atg2 ± WIPI proteins to transfer lipids, the background lipid composition is different with not only DOPC, but also DOPE and DOPS present ( ± PI3P). The authors justify this change by stating that "To stabilize the ATG2A-SUV association, we prepared donor and acceptor SUVs by sonication and included 25% dioleoylphosphatidylethanolamine (DOPE) and 25% dioleoylphosphatidylserine (DOPS) in both vesicles. The role of PE was described above, and DOPS improves the ATG2A-membrane association". My concern is two-fold: First PS is rather low at the ER, where it faces mostly the lumen; second, mammalian lipids generally do not contain two monounsaturated acyl chains and rather show a mixed saturated/unsaturated acyl chain composition. Due to the cumulative effects of lipid-packing defects and high electrostatics, DOPE + DOPS membranes can be super permissive to the binding of many proteins and promote artefactual or, at least, exaggerated adsorption. For example, the yeast lipid transfer protein Osh4 can tether artificially DOPS-rich liposomes (PMID:20008566), despite the fact that this protein does not work as a membrane tether in vivo (for a general discussion on the interest to separate lipid-packing and electrostatics see e.g. PMID:23153485; for a more precise discussion in the case of Osh4 see e.g. Figure 6 in PMID:22162133). Thus, I suspect that the use of more relevant liposomes to mimic ER and ER- derived membranes (e.g. with POPC, POPE and less PS would lead to more clear-cut results. Notably, for

- the modest but clear difference in activity between ATG2A and ATG2A12xD (Figure 3);

- the strange liposome tethering activity of WIPI1 (Figure 4H and Figure 6).

2) It would be important to have an estimate of the efficiency of the lipid transfer reaction (in mol of lipids per mol of protein per unit of time; for example, the lipid transfer protein OSBP, which also combines a tethering mechanism and a lipid transfer activity, transfers one mol of chosterol/OSBP/second; PMID:24209621).

3) Although the assay is limited to the use of fluorescent lipids, it would be informative to assess the selectivity of Atg2A for various lipid classes and species. See recent papers on Vps13.

4) The text in the Results section is sometimes difficult to follow.

4a) When the authors discuss the dose-response effect of ATG2A on the rate and extent of lipid transfer, they invoke interesting parameters such as the off rate of the protein from tethered membrane regions but they do not actually test their hypothesis. With sonicated liposomes at 25 µM lipids, the molar concentration of the liposomes should be in the range of 2 nM (assuming that a spherical liposome of 20 nm in radius has about 15000 lipids). Therefore, even at the lowest protein concentration, most liposomes should be covered by the protein, making the need for protein turnover between liposomes questionable.

4b) It is misleading to state that "ATG2A12×D exhibited lipid transfer activity similar to that of wild-type ATG2A, indicating that ATG2A12×D transfers NBD-PE between the tethered membranes." In fact, the mutant is about 4 times less active than the wild-type form (see also #1 on liposome composition).

4c) Please better explain the experiments shown in Figure 4 and why do they demonstrate that "Thus, ATG2A-mediated lipid transfer in vitro is an exchange reaction."

---

## [Author Response]

Essential revisions:- Test ATG2A lipid transfer using different liposome compositions as indicated by reviewer 3 (point 1).- Include an ATG2 lipid-binding deficient mutant in the lipid transfer assays (reviewer 1, point 1).- Perform additional controls for liposome integrity (reviewer 2 points 1 and 2) and PI3P-dependence in WIPI1 membrane binding.

Thank you for valuable comments. We have completed the first and third points of the essential revisions. As described below, we found that lipid compositions deeply affect lipid transfer, which led us to conclude that lipid-packing defects are essential for lipid transfer (the first point). For the third point, the requested controls yielded expected results. As for the second point, we could not complete the requested mutational analysis, and the reason will be described below. Overall, we have incorporated new data and revised the text significantly in Results and Discussion, which we believe have refined the message of our work. We note that during this revision, two groups published similar stories on ATG2A and yeast Atg2. We have included a paragraph in Discussion to mention these papers. Below is a summary of the changes in figures.

We had seven main and two supplementary figures in the original submission and now have six main and eight supplementary figures. The first supplementary figure in the original version has been partially incorporated into the main Figure 1A (the surface representation of the ATG2A N-terminus homology model showing the cavity). The second supplementary figure that had a speculative structural model for lipid loading/unloading has been removed entirely as a response to reviewer 2’s point 7. The original Figure 6, which was showing WIPI1-induced liposome aggregation, has been moved to Figure 4—figure supplement 4 as we agree with reviewer 2 (point 4). We have taken this opportunity to add new Figure 1E where we present cryo-EM 2D averages of the WIPI4-ATG2A complex that show the predicated cavity in ATG2A. Figure 2 has been updated with larger gel images and a fluorescence picture of the input centrifuge tubes, as requested. We now have two new supplementary figures for new Figure 2 in which we show control DLS and EM results. Figure 3 and Figure 4 has been rearranged significantly to include our new data on lipid transfer with PO lipids. Both figures are supplemented by additional figures showing control DLS and EM data. Figure 5 has been updated to include the WIPI4 pull-down data and show the entire western blots, as requested. The new Figure 6 presents updated models of lipid transfer.

Reviewer #1:Maeda et al. characterize human ATG2A as a lipid transfer protein in vitro. Following up on their previous structural work, they now show that Atg2 is capable of extracting lipids from one membrane and unloading them on to a different one. Furthermore, it is shown that the lipid transfer is greatly enhanced by WIPI1 and WIPI4 under conditions in which one of the vesicles contains PI3P, a result consistent with their previous study. However, in contrast with WIPI4, WIPI1 does not appear to bind directly to ATG2A in solution and appears capable of inducing liposome clustering. Overall, the identification of ATG2A as a lipid transfer protein is a significant advance to the field. Moreover, most the data is of good quality and the experiments are presented in a clear and logical manner. However, I find that a few controls are missing.1) In my opinion the paper would become stronger if an ATG2 lipid-binding deficient mutant was included. I am not sure whether such a mutant is available but if yes, its inclusion would make conclusions stronger.

Since no such mutants of the full-length ATG2A protein were available, we needed to create one. Because the lipid-binding region of the full-length ATG2A is expected to be extensive, impairing lipid binding of the full-length protein would require many mutations. Creating such a mutant protein would be extremely challenging without having the entire structure. With this in mind, we did make an effort into a mutational study. According to our model of lipid transfer (Figure 6B), lipids would move along the long hydrophobic cavity created by the repeats of β-strands, each of which runs roughly perpendicularly to the long axis of the ATG2A rod. Thus, we hypothesized that building a hydrophilic wall across the elongated hydrophobic cavity might stop such movement of lipids and thereby inhibit lipid transfer. To do this, we replaced four hydrophobic residues in a single β-strand in the N-terminus with charged amino acids using a Vps13 structure-based homology model as the guide. We made three constructs, each of which contained four mutations (on β-5, -6, -7) in the N-terminus. Unfortunately, however, we were unable to obtain sufficient amounts of these mutants for biochemical assays (Author Response Image 1). It is likely that the mutations disrupted the folding of the protein. These results give us an impression that it is going to be time-consuming to discover a purifiable full-length ATG2A mutant. Very recently, Valverde et al. reported that two sets of mutations in an isolated ATG2A N-terminal construct impair lipid transfer by this construct. Both mutants contained a massive number of point mutations (eleven and nine) in this just N-terminal construct, suggesting that a mutational study of the full-length would be even more difficult.

**Author response image 1. respfig1:** Expression and purification of the ATG2A full-length mutants. The locations of the mutations are shown on the ATG2A N-terminus homology model. All four point mutations on each β strand were incorporated into one mutant construct (a total of three constructs were tested). SDS-PAGE shows the results of expression of the mutants in Sf9 insect cells and purification with glutathione beads. All three mutants expressed less than the wild-type protein, and little mutant proteins were detected on glutathione beads after wash. The eluted fractions after on-beads TEV digestion contained very little mutant proteins. The poor yields of these mutants prevented biochemical studies on these mutants.

2) The effect on PI3P on ATG2A/WIPI4 binding to membranes has been previously characterized by the authors (Chowdhuri et al., 2018). However, that is not the case for WIPI1. Thus, in face of the data presented, it is not possible to conclude that PI3P plays a critical role in WIPI1 membrane binding as its binding was not tested in the absence of PI3P.

We have performed liposome flotation with WIPI1/WIPI4 and PI3P-free liposomes and now show that WIPI1 also requires PI3P for its membrane binding (Figure 5B; lanes 22–24, subsection “WIPI4/WIPI1 enhance the association of ATG2A with PI3P-containing membranes”). These data are consistent with the previous study by the Hurley group who showed that all four WIPI paralogues do not bind PI3P-free membranes (Baskaran et al., 2012).

3) In my opinion, the results on the cooperativity between WIPI1 and ATG2A are overinterpreted, in particular considering the effect of WIPI1 on liposome morphology and on the ATG2A binding to membranes with packing defects. I wonder whether by altering liposome morphology/curvature, WIPI1 generates membrane packing defects which in turn facilitate ATG2A binding. Perhaps this possibility should be considered and discussed.

Thank you for this thoughtful comment. We agree that this is a possibility, and we now have modified the relevant paragraph with a mention about this case. We still offer the possibility of the protein-protein (WIPI1-ATG2A) interaction as the source of the WIPI1-promoted membrane recruitment of ATG2A, which we think is reasonable because its paralogue WIPI4 interacts with ATG2A, and even a very weak interaction between these two proteins could keep ATG2A on the membrane as it can synergize with ATG2A’s own weak affinity to the membrane. The revised text starts with “The experiments with WIPI1 yielded similar results …”.

Reviewer #2:[…] Even though the manuscript lacks in vivo validation, the data presented by Maeda et al. are an important contribution to the autophagy and provide a crucial biochemical activity for one of the central autophagy factors. As such it will be interesting for the readers of eLife. However, some additional controls should be conducted in order to corroborate the finding that ATG2 is a bona fide lipid transfer protein.1) Figure 2: The authors should test if there are liposomes remaining in the bottom fractions. This could be done for example by electron microscopy or DLS. In addition, the authors should provide a quality control for the integrity of the liposomes containing 20% NBD-PE. Furthermore, the ATG2A input needs to be shown for 2D, I. Also, in 2B it seems there is significantly more overall fluorescence in the tube +ATG2A. Initial levels of fluorescence (input) should be shown.

We have re-performed the extraction assay of Figure 2A and observed the bottom fractions of the samples with and without the ATG2A protein by negative stain EM, as suggested. As shown in Figure 2—figure supplement 2, we were unable to detect any liposomes in the EM specimen of both samples after centrifugation, suggesting that all liposomes floated up in both samples.

To gain insights into “the integrity of the liposomes”, we performed DLS experiments with the liposomes containing 20% NBD-PE in the presence and absence of the ATG2A proteins. As shown in Figure 2—figure supplement 1, these liposomes have an expected size, and ATG2A, which interacts with these liposomes only weakly (Figure 2D), does not change the size of these liposomes. In addition, the negative stain EM micrographs shown in Figure 2—figure supplements 2A and 2C suggest that these liposomes do not undergo morphologic changes upon incubation with ATG2A. Collectively, these results indicate that the integrity of these liposomes is retained in the presence of ATG2A.

We have rerun gels shown in Figure 2D and also 2I with the input samples for each experiment. These gel images are larger than the original images, now covering all gel area from the well to the bottom, as requested by reviewer #3.

We have added the pictures of the tubes before centrifugation, which show that the same amount of NBD-PE liposomes was loaded into each tube. These fluorescent pictures of the tubes were taken using a blue light box and a cell phone camera so that the dynamic range of the picture is poor, and the pixels are saturating particularly at the most intense part (the top of the tube without proteins). These are to help understand the results qualitatively. We generally observe that in the absence of proteins the fluorescent liposomes concentrate at the very top of the top layer but in the presence of proteins, the liposomes are more broadly distributed within the top layer.2) Along similar lines, the authors should test if the addition of ATG2 or ATG2-WIPI4 does disrupt the liposomes used in Figures 2 and 4.

As described above, we have performed negative stain EM and DLS analyses for the liposomes used in Figure 2. The results from these experiments do not seem to suggest that ATG2A disrupts the liposomes. For those liposomes used in Figure 4, we performed DLS analyses as well as cryo-EM imaging of tethering of PI3P-containing liposomes. Both experiments suggest that the ATG2A-WIPI4 complex does not disrupt these liposomes.

3) The authors show in Figure 5 that WIPI1 recruits ATG2 to the membrane but a direct interaction is not detectable (Figure 5A). The authors should show the same pulldown assay for WIPI4. Not only will it give an idea of the ATG2A/WIPI4 interaction strength (that was shown only indirectly as the authors explain in the text) but also will serve as a positive control for the ATG2A/WIPI1 pulldown. In addition, the authors should use more sensitive assays to demonstrate a direct interaction between WIPI1 and ATG2. An alternative explanation would be that WIPI1 disturbs the membrane that these defects are recognized by ATG2.

We have re-performed the affinity pull-down experiment shown in Figure 5A with both WIPI1 and WIPI4. The result confirms that WIPI4 but not WIPI1 interacts with ATG2A under this condition.

Detection of weak interactions by pull-downs is limited by fast dissociation rather than low sensitivity. A workaround is to detect the complex in equilibrium, but it would require substantially higher protein concentrations, which is difficult to achieve with ATG2A. With this in mind, we did try BLI (Octet), which in theory, could detect weak interactions because the detection is done on the surface onto which receptor proteins can be concentrated. We attached ATG2A onto the sensor tip and tried monitoring its interaction with WIPI1 both kinetically and in equilibrium by applying WIPI1 at relatively high concentrations (µM or higher). However, as is often the case with this technique, the experiments suffered from non-specific binding of WIPI1 to the sensor tip despite our efforts to eliminate them. As such, we are not able to offer a conclusive mechanism at this time for the WIPI1-facilitated ATG2A-membrane association. As described above in the responses to reviewer 1, we have included a mention about the alternative possibility that WIPI1 may recruit ATG2A indirectly to the membrane (subsection “WIPI4/WIPI1 enhance the association of ATG2A with PI3P-containing membranes”).

4) The data shown in Figure 6 are somewhat out of context and it is unclear how the liposome clustering activity is integrated into the process of autophagy.

We agree with the reviewer on this point but feel we should provide an explanation as to the suppressed lipid transfer at high concentrations of WIPI1. To improve the readability, we have moved the last paragraph with the original subtitle “WIPI1 clusters PI3P-containing liposomes” to immediately after reporting the results of WIPI1-dependent lipid transfer. And we split the original section titled “WIPI4/WIPI1 facilitate ATG2A-mediated lipid transfer” into two, the first one on WIPI4 and the second on WIPI1. The moved paragraph starts with “In the experiments with WIPI1, we observed that the solutions…”. The original Figure 6 showing WIPI1-induced liposome clustering is now Figure 4—figure supplement 4. We hope these changes improve the readability of the manuscript.

5) In the Introduction, the authors refer to the crystal structure of Vps13 N-terminal part from Chaetomium thermophilum (Introduction, fourth paragraph) described in Kumar et. al mentioning that it can accommodate ~10 lipid molecules, where in fact the number of lipid molecules was estimated for N-terminal domain of Vps13α from *S. cerevisiae* in Mass Spec analysis. This should be clarified in the text.

Thank you for catching this misleading line. We have revised the sentence to cite their work accurately.

6) Figure 1: Adding residues numbers to the cartoon representation of ATG2A would give a better idea of the size of particular domains. The authors should depict the borders of the crystallized homology domain on the cartoon representation of ATG2A.

We have revised Figure 1A to include the residue numbers and indicate domain regions more precisely.

7) Discussion: The discussion about the possible regulation of the lipid loading/unloading appears to be too speculative.

We had felt that we were obligated to offer some insights into the mechanisms given the availability of a homologous structure, but in light of this opinion, we have removed this discussion entirely.

Reviewer #3:

[…] The overall message of the work is novel and gives a key working hypothesis for the mechanism of autophagosome formation. As such, it should be suitable for publication in eLife.A few additional experiments and textual changes might clarify some issues.1) My main concern is about the composition of the liposomes. In Figure 1 where the authors test the ability of Atg2 to extract and deliver lipids, they use a mixture of DOPC and NBD-PE. In contrast, in the figures where they test the ability of Atg2 ± WIPI proteins to transfer lipids, the background lipid composition is different with not only DOPC, but also DOPE and DOPS present ( ± PI3P). The authors justify this change by stating that "To stabilize the ATG2A-SUV association, we prepared donor and acceptor SUVs by sonication and included 25% dioleoylphosphatidylethanolamine (DOPE) and 25% dioleoylphosphatidylserine (DOPS) in both vesicles. The role of PE was described above, and DOPS improves the ATG2A-membrane association". My concern is two-fold: First PS is rather low at the ER, where it faces mostly the lumen; second, mammalian lipids generally do not contain two monounsaturated acyl chains and rather show a mixed saturated/unsaturated acyl chain composition. Due to the cumulative effects of lipid packing defects and high electrostatics, DOPE + DOPS membranes can be super permissive to the binding of many proteins and promote artefactual or, at least, exaggerated adsorption. For example, the yeast lipid transfer protein Osh4 can tether artificially DOPS-rich liposomes (PMID:20008566), despite the fact that this protein does not work as a membrane tetherin vivo (for a general discussion on the interest to separate lipid packing and electrostatics see e.g. PMID:23153485; for a more precise discussion in the case of Osh4 see e.g. Figure 6 in PMID:22162133). Thus, I suspect that the use of more relevant liposomes to mimic ER and ER- derived membranes (e.g. with POPC, POPE and less PS would lead to more clear-cut results. Notably, for- the modest but clear difference in activity between ATG2A and ATG2A12xD (Figure 3);- the strange liposome tethering activity of WIPI1 (Figure 4H and Figure 6).

We are grateful to the reviewer for this valuable and educative comment. We have investigated the effects of negatively charged lipids and the differences between DO and PO lipids-based liposomes on lipid transfer and membrane tethering. Overall, the reviewer’s concern turned out to be completely applicable; the combination of the high PS percentage and the use of DO lipids-based liposomes significantly enhance both membrane tethering and lipid transfer. With the suggested lipid compositions (POPC/POPE and a low (0-5%) POPS), lipid transfer between SUVs still works, albeit slightly slower than 25% DOPS SUVs (Figure 4). However, LUVs (100 nm) with such a PO lipid composition (5% POPS) allowed little lipid transfer (Figure 4G). Similarly, PO LUVs made by extrusion through a 50 nm filter did not allow lipid transfer either (Figure 4H). Because these liposomes are tethered by the ATG2A-WIPI4 complex, lipid transfer itself seems to be inefficient with PO liposomes. Using 5% PI3P/5% POPS-containing donor prepared through a 50 nm filter and 5% POPS-containing acceptor made by sonication, we could demonstrate the stimulation of lipid transfer by WIPI4 and WIPI1 (Figures 4I and 4L), the results consistent with our previous data with 25% DOPS LUVs (Figures 4B and 4J). The suppression of lipid transfer observed at high concentrations of WIPI1 was also reproducible with this new pair (Figure 4L). WIPI1-mediated liposome clustering was also confirmed with PO lipids (Figure 4—figure supplement 4B). Thus, the liposome clustering activity of WIPI1 appears to be independent of lipid-packing defects and negative charges. Interestingly, the difference between the wild-type and the ATG2A^12xD^ mutant proteins were reduced with PO lipids (Figures 3E, 3F, 3I, and 3J), suggesting that the wild-type was rather hyperactive with 25% DOPS liposomes (Figures 3C and 3G). These data imply that ATG2A does not operate at its full capacity in cells, but importantly, ATG2A can mediate lipid transfer under near-physiological conditions.

2) It would be important to have an estimate of the efficiency of the lipid transfer reaction (in mol of lipids per mol of protein per unit of time; for example, the lipid transfer protein OSBP, which also combines a tethering mechanism and a lipid transfer activity, transfers one mol of chosterol/OSBP/second; PMID:24209621).

We have changed the unit of the transfer accordingly and paid more attention to the quantitative aspect of this work. The obtained NBD-PE transfer rate is ~0.017 /protein/second. As mentioned in the second paragraph of the subsection “ATG2A transfers lipids between tethered membranes”, this is comparable to the previously reported rate of NBD-PE transfer by the Mmm1-Mdm12 complex (Kawano et al., 2018). Please note, however, this estimation is not accurate given that the changes in the fluorescence signal in this assay are caused via multiple routes including the transfer of Rh-PE and non-fluorescent lipids. In any case, the obtained rate of ATG2A-mediated lipid transfer seems to be too slow to drive phagophore growth by itself. This is actually not disappointing but rather realizing. As mentioned above in the responses to reviewer 1 and explained below, ATG2A-mediated lipid transfer in vitro is bidirectional. What we are observing in vitro is likely a diffusion-driven dilution of lipids across donors and acceptors. That probably won’t be sufficient to create an autophagosome de novo. This thinking leads us to speculate that the lipid “transport” rate would be much faster during actual autophagosome formation in cells. We will need more factors to fully understand how this would work.

3) Although the assay is limited to the use of fluorescent lipids, it would be informative to assess the selectivity of Atg2A for various lipid classes and species. See recent papers on Vps13.

We have been attempting for years to use the native gel shift assay to demonstrate ATG2A’s lipid binding and investigate lipid specify but have been unsuccessful. As mentioned in Figure 1 legend and in Materials and methods, on our hands, ATG2A aggregates in the wells of native gels, not entering the gels. With this suggestion from reviewer 3, we have revisited to this technique and again tried hard to make this experiment work. However, despite that we tested all the conditions we could think of, none has worked. Meanwhile, Valverde et al., 2019, showed the results asked by the reviewer. We are not sure why we could not make this experiment work. Thus, unfortunately, we are unable to offer insights into lipid specificity.

4) The text in the Results section is sometimes difficult to follow.4a) When the authors discuss the dose-response effect of ATG2A on the rate and extent of lipid transfer, they invoke interesting parameters such as the off rate of the protein from tethered membrane regions but they do not actually test their hypothesis. With sonicated liposomes at 25 µM lipids, the molar concentration of the liposomes should be in the range of 2 nM (assuming that a spherical liposome of 20 nm in radius has about 15000 lipids). Therefore, even at the lowest protein concentration, most liposomes should be covered by the protein, making the need for protein turnover between liposomes questionable.

We thank the reviewer for this thoughtful comment, which led us to re-consider the explanation about the titration experiment. As a result, we have now completely revised this paragraph starting with “We next performed a titration of ATG2A…”. This line of consideration also stimulated us to think about the autophagosome formation more quantitatively, which resulted in revising Discussion.

4b) It is misleading to state that "ATG2A12×D exhibited lipid transfer activity similar to that of wild-type ATG2A, indicating that ATG2A12×D transfers NBD-PE between the tethered membranes." In fact, the mutant is about 4 times less active than the wild-type form (see also #1 on liposome composition).

Thank you for this comment, which we agree with. As mentioned above, our new data with PO lipids suggest that this mutant protein is only slightly less active compared to the wild-type. Thus, it seems that the hydrophobic side of the amphipathic CLR of the wild-type was adding extra strength in binding to DO membranes. Thus, the reason why this mutant would not work in cells is likely due to the impaired localization. Accordingly, the paragraph describing these results, “We also tested a mutant protein, referred to as ATG2A^12×D^, …” has been revised.

*4c) Please better explain the experiments shown in Figure 4 and why do they demonstrate that "Thus, ATG2A-mediated lipid transfer* in vitro *is an exchange reaction."*

We have edited the paragraph including the quoted line for clarity. We now state that ATG2A-mediated lipid transfer is a “bidirectional” reaction instead of “exchange”.